# ATLAS MATTERS: EDGE QUADRATICS FOR CONSISTENT BRAIN CONNECTIVITY PREDICTION

## ABSTRACT

Functional connectivity from resting-state fMRI is a strong substrate for subject-level prediction, yet progress is held back by two issues. First, most architectures ingest FC via node-centric propagation or global attention, leaving higher-order edge interactions implicit. Second, evaluations are inconsistent across seeds, atlas choice, preprocessing, and hyperparameter budgets, which obscures true gains.

We propose a simple edge-image encoder that applies dual atrous spatial pyramid pooling to features and connectivity, coupled with a low-rank quadratic block that makes edge-edge effects explicit and efficient. Beyond design, we introduce a unified protocol with five fixed seeds, harmonized preprocessing, and multiple standard atlases, and we re-run recent GNN and transformer baselines under identical settings. Under this protocol, our model `EdgeQuad` attains the best mean performance on curated functional atlases for ABIDE and ADNI, while on unsupervised parcellations such as Ward and KMeans rankings are mixed, highlighting sensitivity to atlas construction. The quadratic block realizes localized degree-2 interactions with provable stability, explaining robustness. The model is lightweight and computationally efficient. To facilitate rigorous comparison, we release code, exact configs, and per-seed logs via an anonymous link.

## 1 INTRODUCTION

Functional connectivity (FC) with resting-state fMRI represents each subject as a graph whose nodes are regions of interest (ROIs) and whose edges are pairwise Pearson correlations between ROI BOLD signals. This correlation pipeline is standard in large cohorts such as ABIDE and ADNI, where the correlation matrix serves as the adjacency for downstream learning (Martino et al., 2014; Weiner et al., 2015). Building on this representation, models learn from FC using adjacency-as-image CNNs, message-passing GNNs, and graph transformers, often with neuro-inspired pooling or attention (Kawahara et al., 2017; Parisot et al., 2017; Li et al., 2021). These efforts echo principles of brain organization, small-world structure, hubs, and rich-club connectivity (Watts & Strogatz, 1998; Bullmore & Sporns, 2009; van den Heuvel & Sporns, 2011). Recent methods emphasize long-range communication and biologically informed priors (Yu et al., 2024; Peng et al., 2025b), while others question the need for deep propagation via efficient quadratic operators (Yang et al., 2025). Concurrently, dynamic FC studies show that coupling varies over time, so single static snapshots can miss state changes (Hutchison et al., 2013; Preti et al., 2017; Allen et al., 2014).

Despite this progress, two issues hamper trustworthy comparison and deployment of FC-based models. **(i) Reproducibility.** Reported results differ in atlas choice, preprocessing, splits, hyperparameter budgets, and random seeds, making claims hard to compare even on the same dataset. **(ii) Edge modeling.** Most architectures consume the correlation matrix through node-centric propagation or global attention, leaving higher-order edge–edge interactions implicit and often conflating the matrix as both structure and features (Yang et al., 2025).

**Our approach.** We revisit FC modeling with a simple, parameter-light *edge-image* encoder and a *standardized evaluation*. On the modeling side, we treat the FC matrix as an image and extract multi-scale structure before any graph readout: a dual atrous spatial pyramid pooling (ASPP) acts on feature maps and connectivity, and a low-rank quadratic branch makes edge–edge effects explicit while avoiding deep message passing. On the evaluation side, we re-run recent baselines under identical settings (five fixed seeds, fixed splits, harmonized preprocessing, multiple atlas choices), so comparisons reflect architectural merit rather than protocol variance .

Our main contributions are:

- A lightweight **dual-ASPP edge-image model** with a **low-rank quadratic** interaction that exposes edge–edge structure efficiently on FC graphs.

- A **standardized protocol** with aligned seeds, splits, hyperparameter budgets, and common atlas choices, enabling fair and reproducible comparison across methods.

- **Theoretical analysis** showing that the quadratic block realizes rank-$k$ degree-2 interactions localized by dual ASPP and is Lipschitz in both features and refined connectivity, which helps explain robustness under atlas and site variation.

**Benchmark scope and claims.** We position this work as a method plus a rigorous benchmark for FC modeling. We re-implement and evaluate recent high-profile models, including BQN, BioBGT, and ALTER, under the same protocol across AAL, DKT, and Schaefer atlases at multiple resolutions. *On curated functional or anatomical atlases, our method attains the best mean performance under seed-controlled evaluation. On unsupervised parcellations such as Ward and KMeans, rankings are mixed and margins are smaller.* These findings underscore the impact of atlas choice on reported gains. We release code, exact configs, and per-seed logs at an anonymous link for full reproducibility.

The rest of the paper is organized as follows. In the next section, present related work, and background material for FC graphs. In Section 3, we introduce the EdgeQuad architecture and its components, Sections 4.1 and 4.3 describe the datasets, unified evaluation protocol (Appendix B.4), and main results, and Sec. 4.4 presents ablations and theoretical connections. We next introduce the essential background and notation required to present the model and evaluation protocol.

## 2 BACKGROUND

### 2.1 RELATED WORK

**Graph learning on brain networks.** A standard pipeline converts resting-state fMRI into a subject-level network whose nodes are ROIs and whose edge weights are Pearson correlations between ROI BOLD time series. Learning typically treats the connectivity matrix either as an image for convolutional models (e.g., BrainNetCNN) or as an adjacency for GNNs with task-driven pooling and interpretability modules (Kawahara et al., 2017; Parisot et al., 2017; Li et al., 2021). Transformer variants have recently reported strong results: Brain Network Transformer introduces connection-profile features and an orthonormal clustering readout (Kan et al., 2022); residual and gated designs target ASD and cognitive traits (Wang et al., 2024; Qu et al., 2024). Brain-specific graph transformers encode long-range communication via biased random walks (Yu et al., 2024) and incorporate small-world/module priors through biologically informed attention, including joint structural–functional modeling (Peng et al., 2025b;a). Standardized benchmarks (BrainGB) have begun to improve comparability (Cui et al., 2023), but evaluations still vary widely in atlas choice, splits, seeds, and hyperparameter budgets.

Beyond node-centric architectures, several recent works operate directly on edge signals or higher-order topological structure. Park et al. (2023) convolve *directed* graph edges via the Hodge Laplacian for brain network analysis; Fuchsgruber et al. (2025) study GNNs for general edge signals with orientation equivariance and invariance; and Lecha et al. (2025) introduce directed simplicial neural networks that act on higher-order simplices. These approaches highlight the importance of edge- and motif-level information, but are formulated for directed or simplicial structures with explicit incidence information, whereas rs-fMRI FC in our setting yields dense, undirected correlation matrices without such auxiliary structure.

Several recent works focus on hierarchical pooling, contrastive objectives, or multi-atlas distillation rather than fixed-atlas supervised FC modeling. BrainGNN learns task-specific node selection and hierarchical pooling with sparsity and interpretability losses on ROI masks (Li et al., 2021). Contrastive pooling methods for brain graphs (Xu et al., 2024; Tang et al., 2024a) build multi-level coarsened summaries and are trained with contrastive or auxiliary objectives. Multi-atlas distillation approaches (Xu et al., 2025) combine several parcellations simultaneously through atlas-specific subnetworks and consistency losses, which differs from our single-atlas, fixed-graph setting.

**Dynamic FC and spatio-temporal models.**  A parallel line addresses nonstationarity in BOLD by estimating time-varying graphs or using dynamic graph transformers and hybrid spatio-temporal models (Hutchison et al., 2013; Preti et al., 2017; Allen et al., 2014; Campbell et al., 2024; Guan et al., 2024; Shehzad et al., 2025; Tang et al., 2024b). While these approaches improve temporal sensitivity, most rs-fMRI graph classification pipelines still consume correlations through node-centric propagation or attention, so higher-order edge–edge interactions are typically left implicit in the learned node representations, with edge- and simplicial-based architectures (Park et al., 2023; Fuchsgruber et al., 2025; Lecha et al., 2025) as notable exceptions.

Our work differs in two ways within this rs-fMRI setting. (i) *Modeling:* we treat the FC matrix as an *edge image* and make edge–edge effects explicit via a lightweight dual-ASPP encoder and a *low-rank quadratic* interaction branch, avoiding deep message passing and keeping parameters small. In contrast to the above edge-/simplicial-based models, our design is tailored to dense, undirected Pearson-FC graphs and does not require constructing additional incidence or simplicial complexes. (ii) *Evaluation:* we re-run recent high-profile baselines under a single protocol (aligned seeds/splits, harmonized preprocessing, and all common atlases), isolating architectural merit from protocol variance and providing a standardized testbed that future edge- and simplicial-based models can also be evaluated on.

## 2.2 PRELIMINARIES FOR FC GRAPHS

**From fMRI to brain networks.**  For readers new to neuroimaging, we include an ML-friendly primer on how resting-state fMRI is converted into a subject-level FC graph (atlas, preprocessing, correlation, and graph construction) in Section A.

**Notation.**  A subject's brain network is denoted by $\mathcal{G} = (\mathcal{V}, \mathbb{E}, \mathcal{X})$ with $N = |\mathcal{V}|$ is the number of ROIs defined by the chosen atlas. The connectivity (adjacency) is $\mathcal{A} = [a_{ij}] \in \mathbb{R}^{N \times N}$ (typically Pearson correlations), and node features are $\mathcal{X} \in \mathbb{R}^{N \times D}$ with row $\mathbf{x}_i$. We write $\tilde{\mathcal{A}} = \mathcal{A} + \mathbf{I}$ and $\tilde{\mathcal{D}} = \mathrm{diag}(\tilde{d}_1, \ldots, \tilde{d}_N)$ with $\tilde{d}_i = \sum_j \tilde{a}_{ij}$. Given graphs $\{\mathcal{G}^{(i)}\}_{i=1}^L$ and labels $\{y^{(i)}\}_{i=1}^L$, the goal is to predict $\hat{y}^{(i)} = f(h(\mathcal{G}^{(i)}))$ by minimizing the empirical loss $\frac{1}{L} \sum_i \ell(\hat{y}^{(i)}, y^{(i)})$.

**GNNs and Transformers.**  Message passing updates node embeddings by aggregating neighbors then combining with the center node; a common instance is the renormalized graph convolution

$$H^{(\ell)} = \sigma\big(\tilde{\mathcal{D}}^{-1/2} \tilde{\mathcal{A}} \tilde{\mathcal{D}}^{-1/2} H^{(\ell-1)} W^{(\ell)}\big),$$

where $\mathbf{H}^{(\ell)} \in \mathbb{R}^{N \times F_\ell}$ is the matrix of node embeddings at layer $\ell$, $\mathbf{W}^{(\ell)}$ is a trainable weight matrix, and $\tilde{\mathbf{A}}$ and $\tilde{\mathbf{D}}$ are the adjacency and degree matrices with self-loops. The resulting node embeddings are then aggregated using graph-level pooling and passed to an MLP for classification. Transformers aggregate globally with self-attention. For tokens $Z \in \mathbb{R}^{N \times F}$,

$$Q = ZW_Q, \quad K = ZW_K, \quad V = ZW_V, \quad \mathrm{Attn}(Z) = \mathrm{softmax}\big(QK^\top / \sqrt{F}\big)V,$$

with structure injected via attention masks/biases derived from $\mathcal{A}$ or via graph-aware positional encodings (e.g., Laplacian eigenvectors, random-walk features).

**Limits for FC graphs and our design choices.**  In FC, explicit node attributes are scarce. In standard FC benchmarks, the model input at training and inference is essentially a static FC matrix $C$ plus minimal per-ROI metadata, so the rich node signals that generate $C$ (time series, task contrasts, parcel-wise features) are not available at inference and the primary informative signal resides on edges rather than independent node features. Many pipelines reuse rows of $A$ or degree statistics as features, which makes propagation largely a mixing of powers of $A$ and can add little new information. Deeper stacks also risk over-smoothing and over-squashing at modest cohort sizes. FC is inherently *edge-valued*, yet standard layers propagate *node* states and leave edge–edge relations implicit. These observations motivate our choices: (i) decouple structure from attributes and elevate edges to first-class citizens by treating the correlation matrix as an *edge image*; (ii) capture multi-scale organization with dual ASPP applied to feature maps and to connectivity; (iii) model edge–edge effects explicitly with a *low-rank quadratic* interaction that stays efficient on dense FC. Empirically and theoretically we find this yields stable, competitive performance under a standardized evaluation across seeds and atlas choices on curated functional atlases, while rankings are mixed on unsupervised parcellations, which we report transparently.

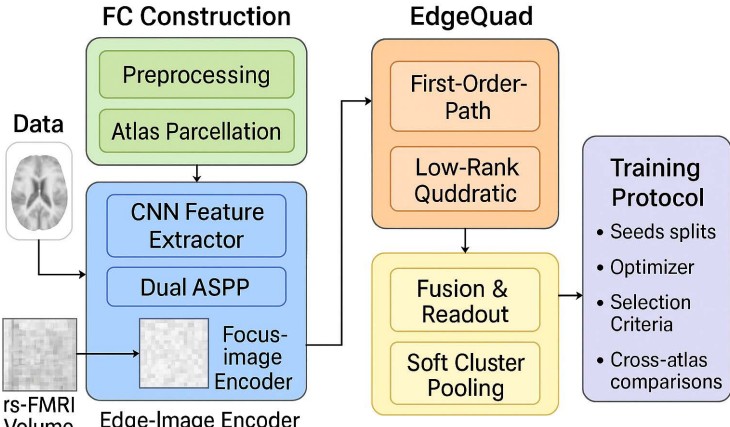

Figure 1: **Conceptual overview of EdgeQuad.** rs-fMRI volumes are preprocessed and parcellated to form FC matrices ("FC Construction"). The FC matrix is treated as an edge image and encoded by a CNN with dual ASPP ("Edge Image Encoder"). EdgeQuad combines a first order path and a low rank quadratic branch, then applies fusion and soft clustering to obtain graph embeddings. All models are trained under a unified protocol with fixed splits, seeds, and cross atlas comparisons. Detailed flowchart is given in the Appendix (Figure 3).

With these preliminaries in place, we now describe EdgeQuad in a single forward pass from inputs to outputs.

## 3 EdgeQuad: Edge-Quadratics for Functional Connectivity

This section introduces the components of EdgeQuad in the order they are used at inference. Figures 1–2 provide a visual guide. We model the functional connectivity (FC) matrix directly as an *edge image* and build a lightweight encoder that exposes multi-scale organization before any graph readout. We preserve the atlas ordering of ROIs, so convolutional kernels scan FC patches that follow anatomical and functional groupings, and are therefore encouraged to detect within-network and between-network motifs rather than arbitrary pixel neighborhoods. A CNN followed by dual atrous spatial pyramid pooling (ASPP), one branch on feature maps, one directly on connectivity, yields node embeddings and a refined connectivity that captures both local modules and long-range integration. We then introduce a *low-rank quadratic* interaction that makes edge–edge effects explicit and fuse it with a first-order path via a content gate and cluster pooling. The design decouples structure from attributes, avoids deep message passing, and is trained end-to-end from scratch for site- and atlas-robust performance. The full architecture is illustrated in Figure 1.

### 3.1 Dual-ASPP Edge Imaging with Low-Rank Quadratic Interactions

**Problem and brain-network view.** Functional brain networks are modeled as graphs $G = (V, E)$, where $|V| = N$ denotes cortical or subcortical regions and weighted edges represent functional connectivity (FC), typically estimated from Pearson correlations or coherence (see Section A). FC reorganizes across tasks, development, and disease, modules, hubs, and long-range integration shift in ways that simple linear deformations cannot capture (Bertolero et al., 2015; Fornito et al., 2015). We now describe how EdgeQuad encodes these FC graphs by treating $C$ as an edge image and applying a fully convolutional encoder with ASPP, where 2D convolutions operate on atlas-ordered FC matrices and are intended to exploit network-level locality (modules, hub motifs, and inter-system bridges) rather than voxel-level spatial locality.

**ASPP-refined quadratic interactions on brain graphs.** We refine connectivity with a second ASPP:

$$C' = \mathcal{A}_{\text{edge}}(C), \qquad S = \hat{C}'X, \quad \hat{C}' = D^{-1/2}C'D^{-1/2}, \tag{1}$$

where $\mathcal{A}_{\text{edge}}$ aggregates neighborhoods at multiple dilation rates to capture local modules and long-range integration; degree normalization stabilizes density and site variability (see Figure 7 for a subject-level example of how dual ASPP refines FC).

**Edge affinity.** Given the refined FC $C' \in \mathbb{R}^{N \times N}$, we define a symmetric, degree-normalized edge affinity

$$D_{ii} = \sum_j |C'_{ij}|, \quad S = \tfrac{1}{2}\big(C' + C'^{\top}\big), \quad A_{\text{edge}} = D^{-\frac{1}{2}} |S| D^{-\frac{1}{2}}.$$

This yields a stable normalization for signed graphs; using only the positive part of $C'$ gave similar results in our experiments.

**Feature affinity.** Let $F' \in \mathbb{R}^{N \times d_f}$ be the node features after the feature-ASPP branch. We build a content-based affinity

$$S_{\text{feat}} = \frac{(F'W_q)(F'W_k)^{\top}}{\sqrt{d_a}}, \qquad A_{\text{feat}} = \text{softmax}_{\text{row}}(S_{\text{feat}}),$$

optionally sparsified to $k$-NN for efficiency. Thus $A_{\text{edge}}$ is derived from $C'$, while $A_{\text{feat}}$ is learned from $F'$, and they provide complementary signals for propagation.

A first-order update is

$$B = XW_C, \qquad W_C \in \mathbb{R}^{d' \times d'}. \tag{2}$$

To make interaction effects explicit, we add a *quadratic branch*. For node $i$:  $Q_i = W_Q\big(x_i \odot S_i\big)$, where $x_i$ is the $i$-th row of $X$ and $W_Q \in \mathbb{R}^{d' \times d'}$ gates $x_i$ by its ASPP-refined neighborhood summary $S_i$.

**Low-rank efficiency and implementation.** To control compute, parameterize $W_Q$ in low rank. In batched form for $X \in \mathbb{R}^{B \times N \times d'}$,

$$\text{Quad}(X) = \Big((XV^{\top}) \odot \big((\hat{C}'X)U^{\top}\big)\Big) W_o, \tag{3}$$

with $U, V, W_o \in \mathbb{R}^{k \times d'}$ and $k \ll d'$. This uses two thin projections and a pointwise product; $Q = \text{Quad}(X) \in \mathbb{R}^{B \times N \times d'}$.

**Gated fusion and objective.** We now combine the first-order path $B$ and the quadratic path $Q$ into a single node representation that will be pooled at the graph level. We fuse first- and second-order effects through a content gate:

$$Y_i = \sigma(x_i W_g) \odot \big(B_i + \alpha Q_i\big), \qquad W_g \in \mathbb{R}^{d' \times d'}, \ \alpha \in \mathbb{R}. \tag{4}$$

For graph-level readout we use clustering-based pooling with soft assignments $\Pi \in \mathbb{R}^{N \times K}$,

$$Z = \Pi^{\top} Y \in \mathbb{R}^{K \times d'}, \tag{5}$$

and predict $\hat{y} = g_\varphi(Z)$. We train *all* parameters (CNN, both ASPPs, quadratic branch, gate, pooling, head) from scratch by minimizing

$$\min_{\Theta} \mathcal{L}(\hat{y}, y), \quad \Theta = \{\theta, \mathcal{A}_{\text{feat}}, \mathcal{A}_{\text{edge}}, W_C, U, V, W_o, W_g, \alpha, \Pi, \varphi\}. \tag{6}$$

**Compute/budget notes.** Now, we summarize the computational footprint of the quadratic head and motivate our choice of rank and placement. Let $n = d'$ and spatial size $(H', W')$. The incremental cost of equation 3 per layer scales as $\mathcal{O}(knN)$ (node view) or $\mathcal{O}(knH'W')$ (feature-map view), which is small for $k \ll n$. We place quadratic blocks at *high receptive field* stages and set $k \in \{4, 8, 16\}$ for a strong accuracy–efficiency trade-off.

**Novelty and positioning.** Putting these components together, we can now situate EdgeQuad relative to existing FC encoders and graph models. We (i) treat FC as an *edge image* and apply dual ASPP, $\mathcal{A}_{\text{feat}}$ on features and $\mathcal{A}_{\text{edge}}$ on connectivity, to expose multi-scale modular structure and long-range integration before any graph readout; (ii) make *edge–edge effects explicit* via a low-rank quadratic branch conditioned on $\hat{C}'X$; and (iii) *decouple* structure from attributes and fuse first- and second-order signals with a content gate and atlas-respecting cluster pooling. Unlike message passing or quadratic-only models, this combination yields a simple, train-from-scratch alternative that is efficient and robust under standardized evaluation.

## 3.2 THEORETICAL ANALYSIS

While our empirical results already demonstrate strong performance, it is important to clarify what the quadratic branch contributes beyond heuristics. We therefore provide a compact theoretical analysis that addresses three questions: (i) what kind of degree–2 interactions the low-rank quadratic block can represent, (ii) how dual ASPP confines these interactions to multi-scale neighborhoods, and (iii) how the resulting map behaves under perturbations of embeddings and connectivity.

**Setup.** Let $C \in \mathbb{R}^{N \times N}$ denote the functional connectivity. A CNN+ASPP on the edge image produces feature maps $F' = \mathcal{A}_{\text{feat}}(\Phi_\theta(C))$, which are pooled to node embeddings $X \in \mathbb{R}^{N \times d'}$. A second ASPP on connectivity yields $C' = \mathcal{A}_{\text{edge}}(C)$ and a neighbor summary $S = C'X$ (optionally degree-normalized with $\hat{C}' = D^{-1/2}C'D^{-1/2}$). The quadratic block uses a low-rank interaction

$$Q = \left( (XV^\top) \odot ((C'X)U^\top) \right) W_o, \qquad U, V, W_o \in \mathbb{R}^{k \times d'}, \; k \ll d', \tag{7}$$

and the fused update is $Y_i = \sigma(x_i W_g) \odot (B_i + \alpha Q_i)$ with $B = XW_C$.

**Representation of degree–2 interactions.** The first result characterizes the quadratic branch as an explicit degree–2 interaction between a node and its ASPP-refined neighborhood:

**Proposition 1** (Factorized degree–2 lifting). *For fixed $U, V, W_o$, each output coordinate $Q_i(t)$ can be written as*

$$Q_i(t) = \sum_{p=1}^{k} \langle \tilde{v}_{p,t}, x_i \rangle \cdot \left\langle \tilde{u}_{p,t}, \sum_{j=1}^{N} C'_{ij} x_j \right\rangle,$$

*for suitable $\tilde{u}_{p,t}, \tilde{v}_{p,t} \in \mathbb{R}^{d'}$. Thus $Q$ realizes a rank-$k$ degree–2 polynomial in node features with coefficients linear in the ASPP-refined connectivity $C'$.*

If the encoder from $C$ to $X$ is locally piecewise linear, this extends to the original FC matrix:

**Proposition 2** (From edge image to degree–2 polynomials in $C$). *If the encoder from $C$ to $X$ is locally (piecewise) linear, then each $x_i$ is a linear functional of local patches of $C$. Hence, each $Q_i(t)$ in Prop. 1 is a degree–2 polynomial in entries of $C$ supported on the union of the receptive fields of $\mathcal{A}_{\text{feat}}$ and $\mathcal{A}_{\text{edge}}$.*

Together, Propositions 1 and 2 show that the quadratic block implements explicit degree–2 (edge–edge) terms that are localized by the dual ASPP receptive fields instead of being approximated implicitly by deep message passing.

**Rank and efficiency.** The next result formalizes the expressivity–efficiency tradeoff in the rank parameter $k$:

**Proposition 3** (Rank–$k$ expressivity). *Let $z_i = [x_i; (C'X)_i] \in \mathbb{R}^{2d'}$. The class realized by equation 7 equals the set of cross-block quadratic forms*

$$Q_i(t) = z_i^\top M_t z_i, \qquad M_t = \begin{bmatrix} 0 & R_t \\ 0 & 0 \end{bmatrix},$$

*whose off-diagonal block $R_t \in \mathbb{R}^{d' \times d'}$ has rank at most $k$. As $k$ increases, this class monotonically approaches dense cross-block quadratics while using $\mathcal{O}(kd')$ parameters.*

This predicts the empirical pattern we observe in the rank sweep: increasing $k$ improves accuracy up to saturation, after which larger ranks mainly add cost.

**Stability.** Finally, we bound the sensitivity of the quadratic branch to perturbations in $X$ and $C'$:

**Proposition 4** (Lipschitz stability). *Assume $\|X\|_F \leq M_x$, $\|C'\|_2 \leq \rho$, and $\|U\|_2, \|V\|_2, \|W_o\|_2 \leq M$. Then for perturbations $(\Delta X, \Delta C')$,*

$$\|\text{Quad}(X + \Delta X) - \text{Quad}(X)\|_F \leq M^2 \sqrt{k} \left( (1+\rho)\|\Delta X\|_F + M_x \|\Delta C'\|_2 \right).$$

*Hence, the quadratic branch is Lipschitz in both $X$ and $C'$, with constants controlled by the low-rank factors and the spectrum of $C'$.*

Degree normalization of $C'$ and moderate $k$ therefore improve conditioning and robustness, which we exploit in our design. Proofs of all propositions and a short discussion of design choices are provided in Appendix D.

## 4 EXPERIMENTAL SETUP

### 4.1 DATASETS AND PREPROCESSING

We evaluate on four public rs-fMRI cohorts: ABIDE for autism vs control (Martino et al., 2014), ADNI for Alzheimer's vs control (Weiner et al., 2015), PPMI for Parkinson's disease (Marek et al., 2011), and ADHD200 for ADHD vs control (The ADHD-200 Consortium, 2012) (See Table 1). All datasets are organized in BIDS format and preprocessed with fMRIPrep using its default workflow (Xu et al., 2023). For each parcellation in Table 8, ROI time series are obtained by voxel averaging, followed by regression of standard nuisance terms including motion parameters and tissue signals (Power et al., 2014). Functional connectivity is computed with pairwise Pearson correlation on the denoised time series, and we retain fully weighted signed FC matrices (Smith et al., 2011). Each subject yields a brain network represented as a weighted adjacency $C \in \mathbb{R}^{N \times N}$. Subjects failing basic QC are excluded. These preprocessed FC graphs form the input to all models under a single shared implementation and evaluation setup described next.

Table 1: Cohort summary

| Dataset | Condition | #Subjects | Split | Task |
|---------|-----------|-----------|-------|------|
| ABIDE | Autism | 1025 | 537/488 | Binary |
| ADNI | Alzheimer's | 138 | 80/58 | Binary |
| PPMI | Parkinson's | 195 | 15/113/14/53 | 4-class |
| ADHD200 | ADHD | 459 | 229/230 | Binary |

### 4.2 IMPLEMENTATION DETAILS

**Setup.** All experiments run on a single GPU node (Ganymede cluster) with PyTorch. For each dataset we perform **random subject splits** with a fixed ratio of **70%/10%/20%** for train/val/test. We repeat the full pipeline over **five** independent runs with seeds {42, 1042, 2042, 3042, 4042}; results are reported as the mean across runs and standard deviation.

**Training protocol.** Models are trained for **200 epochs** with **batch size 16**, using **Adam** (initial learning rate $1 \times 10^{-4}$, weight decay $1 \times 10^{-4}$). The learning rate is decayed during training toward a floor of $1 \times 10^{-5}$. Unless a method requires a specific nonlinearity, the activation is selected from {GELU, LeakyReLU, ELU} based on validation performance. Depth (number of blocks/layers $k$) is chosen from {1,2,3,4,5}; dropout is selected from {0.0, 0.1, 0.2, 0.3}. **Model selection** is performed by the *minimum validation loss* checkpoint for each seed; all test metrics are computed from that checkpoint. To ensure fairness across baselines (GNN/Transformer/CNN variants), we use the same data splits, optimizer settings, and hyperparameter grids whenever applicable, differing only where a method's architecture mandates it. We summarize hyperparameters and other details in Table 7. Empirical runtime, throughput, memory, and FLOP comparisons for all models on ABIDE AAL116 are reported in Appendix B.3 (Table 12). Our implementation and code is available at the link [1].

**Evaluation Protocol.** We propose a unified, fair evaluation protocol for brain-network classification that eliminates inconsistent and non-comparable practices: (1) adopt a **7:1:2 stratified train/validation/test split** to preserve class ratios across subsets; (2) run each experiment with 5 independent random seeds to avoid single-seed variance; (3) train for up to 200 epochs and select the best epoch using a dataset-appropriate criterion, **macro AUC (one-vs-rest)** for PPMI (multi-class, highly imbalanced) and **minimum validation loss** for ABIDE/ADHD (binary); (4) report test-set **mean ± std** for the chosen selection metric along with complementary metrics, PPMI: macro AUC, macro recall, macro specificity, overall accuracy; ABIDE/ADHD: AUC, sensitivity, specificity; and (5) fix and publish the split indices and seeds to ensure exact reproducibility. This protocol addresses the major flaws of prior evaluations that (a) cherry-pick the single highest AUC without a consistent selection rule, (b) rely on a single seed, (c) ignore class imbalance by using accuracy or binary AUC on multi-class data, (d) use arbitrary or opaque splits (risking distribution shift or leakage), and (e) omit dispersion statistics, all of which make "meaningful comparison" across models impossible (See Appendix B.4). By standardizing splits, selection criteria, and reporting, our protocol yields robust, transparent, and directly comparable results for future models. See Table 7 for a summary.

---

[1] https://anonymous.4open.science/r/EDGEQUADD-531F

Table 2: **Results.** We benchmark all methods under a unified protocol (aligned splits, preprocessing, and hyperparameter grids) across five atlases. For ABIDE and ADNI (binary), we report mean±std AUC/ACC over 5 seeds. For PPMI (4-class), we report Macro AUC/Accuracy. Further metrics are provided in Table 9.

**ABIDE (Binary)**

| | AAL116 | | SCHAEFER100 | | HARVARD48 | | WARD100 | | KMEANS100 | |
|---|---|---|---|---|---|---|---|---|---|---|
| Method | AUC | ACC | AUC | ACC | AUC | ACC | AUC | ACC | AUC | ACC |
| GCN | $64.51_{\pm2.58}$ | $60.39_{\pm3.43}$ | $64.75_{\pm3.10}$ | $60.98_{\pm3.79}$ | $63.85_{\pm2.12}$ | $60.39_{\pm2.29}$ | $50.10_{\pm3.49}$ | $49.61_{\pm2.60}$ | $54.08_{\pm3.48}$ | $\mathbf{53.53_{\pm3.65}}$ |
| GPS | $60.93_{\pm2.05}$ | $58.82_{\pm2.48}$ | $63.37_{\pm3.55}$ | $58.63_{\pm3.84}$ | $62.80_{\pm1.43}$ | $58.04_{\pm1.90}$ | $49.86_{\pm4.47}$ | $50.98_{\pm2.56}$ | $51.94_{\pm3.96}$ | $51.57_{\pm3.85}$ |
| BrainNet | $66.80_{\pm2.12}$ | $62.16_{\pm2.43}$ | $67.55_{\pm1.34}$ | $64.31_{\pm1.37}$ | $62.68_{\pm3.33}$ | $58.53_{\pm2.27}$ | $49.75_{\pm2.27}$ | $50.49_{\pm1.89}$ | $47.88_{\pm2.49}$ | $48.43_{\pm2.64}$ |
| BioBGT | $56.55_{\pm2.93}$ | $55.88_{\pm2.99}$ | $59.28_{\pm1.74}$ | $53.92_{\pm2.18}$ | $55.28_{\pm2.29}$ | $51.96_{\pm1.15}$ | $\mathbf{60.11_{\pm4.69}}$ | $\mathbf{55.88_{\pm3.01}}$ | $47.57_{\pm2.07}$ | $48.04_{\pm1.49}$ |
| ALTER | $69.22_{\pm1.66}$ | $58.04_{\pm3.64}$ | $68.40_{\pm1.10}$ | $64.12_{\pm1.18}$ | $63.93_{\pm1.22}$ | $57.84_{\pm1.52}$ | $50.97_{\pm1.89}$ | $48.43_{\pm1.00}$ | $48.84_{\pm0.97}$ | $51.57_{\pm2.29}$ |
| BQN | $68.59_{\pm1.48}$ | $63.35_{\pm3.22}$ | $71.36_{\pm0.73}$ | $63.83_{\pm0.69}$ | $70.25_{\pm1.31}$ | $62.97_{\pm1.63}$ | $55.87_{\pm5.07}$ | $52.10_{\pm3.93}$ | $\mathbf{54.30_{\pm2.85}}$ | $50.74_{\pm2.16}$ |
| EdgeQuad | $\mathbf{70.60_{\pm0.93}}$ | $\mathbf{63.86_{\pm0.61}}$ | $\mathbf{72.92_{\pm0.76}}$ | $\mathbf{65.96_{\pm2.21}}$ | $\mathbf{71.18_{\pm1.17}}$ | $\mathbf{66.51_{\pm2.20}}$ | $52.54_{\pm5.14}$ | $51.10_{\pm2.42}$ | $53.33_{\pm7.04}$ | $53.45_{\pm1.12}$ |

**ADNI (Binary)**

| | AAL116 | | SCHAEFER100 | | HARVARD48 | | WARD100 | | KMEANS100 | |
|---|---|---|---|---|---|---|---|---|---|---|
| Method | AUC | ACC | AUC | ACC | AUC | ACC | AUC | ACC | AUC | ACC |
| GCN | $88.50_{\pm6.63}$ | $\mathbf{81.54_{\pm6.15}}$ | $80.50_{\pm6.96}$ | $69.23_{\pm6.88}$ | $71.00_{\pm5.83}$ | $66.15_{\pm6.15}$ | $52.50_{\pm5.24}$ | $30.00_{\pm12.75}$ | $55.50_{\pm10.30}$ | $50.77_{\pm3.77}$ |
| GPS | $83.50_{\pm4.06}$ | $73.85_{\pm3.77}$ | $67.00_{\pm3.67}$ | $61.54_{\pm6.88}$ | $57.00_{\pm3.32}$ | $49.23_{\pm3.77}$ | $49.00_{\pm6.04}$ | $35.38_{\pm3.77}$ | $52.50_{\pm12.04}$ | $43.08_{\pm10.43}$ |
| BrainNet | $85.50_{\pm3.32}$ | $76.15_{\pm3.77}$ | $74.00_{\pm2.55}$ | $73.08_{\pm3.44}$ | $73.50_{\pm6.63}$ | $66.92_{\pm6.71}$ | $56.50_{\pm14.88}$ | $52.31_{\pm8.28}$ | $57.50_{\pm11.51}$ | $50.77_{\pm14.47}$ |
| BioBGT | $71.43_{\pm15.76}$ | $61.46_{\pm14.27}$ | $81.90_{\pm8.38}$ | $61.54_{\pm4.87}$ | $83.33_{\pm8.42}$ | $\mathbf{76.92_{\pm8.97}}$ | $\mathbf{71.43_{\pm12.87}}$ | $\mathbf{58.46_{\pm12.50}}$ | $57.62_{\pm9.57}$ | $52.31_{\pm12.31}$ |
| ALTER | $85.00_{\pm1.58}$ | $\mathbf{81.54_{\pm3.77}}$ | $75.00_{\pm1.58}$ | $73.85_{\pm3.77}$ | $76.00_{\pm2.55}$ | $70.77_{\pm3.08}$ | $62.50_{\pm8.22}$ | $49.23_{\pm10.43}$ | $69.00_{\pm6.63}$ | $47.69_{\pm7.54}$ |
| BQN | $93.53_{\pm1.00}$ | $80.69_{\pm2.76}$ | $94.12_{\pm2.03}$ | $\mathbf{78.62_{\pm2.58}}$ | $85.69_{\pm3.91}$ | $75.17_{\pm7.98}$ | $60.10_{\pm8.18}$ | $55.17_{\pm6.90}$ | $68.33_{\pm5.38}$ | $59.28_{\pm4.57}$ |
| EdgeQuad | $\mathbf{96.76_{\pm0.73}}$ | $80.96_{\pm6.42}$ | $\mathbf{95.39_{\pm1.85}}$ | $76.11_{\pm2.97}$ | $\mathbf{91.37_{\pm4.21}}$ | $65.91_{\pm1.21}$ | $58.76_{\pm8.35}$ | $56.73_{\pm6.74}$ | $\mathbf{69.71_{\pm13.49}}$ | $\mathbf{61.01_{\pm1.72}}$ |

**PPMI (4-class)**

| | AAL116 | | SCHAEFER100 | | HARVARD48 | | WARD100 | | KMEANS100 | |
|---|---|---|---|---|---|---|---|---|---|---|
| Method | AUC | ACC | AUC | ACC | AUC | ACC | AUC | ACC | AUC | ACC |
| GCN | $60.66_{\pm6.47}$ | $57.89_{\pm3.33}$ | $\mathbf{67.11_{\pm4.03}}$ | $51.58_{\pm6.14}$ | $\mathbf{59.91_{\pm4.67}}$ | $58.95_{\pm3.94}$ | $\mathbf{55.62_{\pm4.60}}$ | $\mathbf{60.00_{\pm6.32}}$ | $54.63_{\pm9.88}$ | $51.58_{\pm2.11}$ |
| GPS | $60.93_{\pm2.05}$ | $\mathbf{58.82_{\pm2.48}}$ | $65.88_{\pm2.42}$ | $40.00_{\pm5.37}$ | $57.82_{\pm3.26}$ | $45.26_{\pm4.21}$ | $54.12_{\pm11.87}$ | $40.00_{\pm4.21}$ | $53.69_{\pm12.36}$ | $50.53_{\pm12.72}$ |
| BrainNet | $62.44_{\pm8.62}$ | $53.68_{\pm6.98}$ | $62.32_{\pm4.18}$ | $50.00_{\pm3.72}$ | $53.27_{\pm3.10}$ | $52.63_{\pm6.86}$ | $45.58_{\pm10.79}$ | $46.32_{\pm4.88}$ | $30.78_{\pm8.36}$ | $39.47_{\pm6.00}$ |
| BioBGT | $47.26_{\pm16.48}$ | $57.89_{\pm0.00}$ | $52.13_{\pm5.14}$ | $\mathbf{57.89_{\pm0.00}}$ | $45.57_{\pm5.70}$ | $57.89_{\pm0.00}$ | $55.42_{\pm12.35}$ | $57.89_{\pm0.00}$ | $48.27_{\pm7.42}$ | $\mathbf{57.89_{\pm0.00}}$ |
| ALTER | $61.97_{\pm5.48}$ | $52.63_{\pm2.43}$ | $61.56_{\pm1.51}$ | $51.58_{\pm6.14}$ | $55.79_{\pm7.26}$ | $\mathbf{61.05_{\pm5.37}}$ | $41.45_{\pm7.60}$ | $55.79_{\pm9.76}$ | $57.60_{\pm1.89}$ | $54.74_{\pm2.58}$ |
| BQN | $50.00_{\pm0.00}$ | $21.67_{\pm20.52}$ | $50.00_{\pm0.00}$ | $31.67_{\pm24.45}$ | $50.00_{\pm0.00}$ | $22.92_{\pm3.23}$ | $50.00_{\pm0.00}$ | $24.58_{\pm19.16}$ | $50.00_{\pm0.00}$ | $18.33_{\pm21.06}$ |
| EdgeQuad | $\mathbf{66.34_{\pm2.17}}$ | $49.17_{\pm1.67}$ | $59.63_{\pm4.05}$ | $47.50_{\pm5.00}$ | $58.69_{\pm5.12}$ | $53.33_{\pm7.52}$ | $45.04_{\pm4.32}$ | $55.83_{\pm4.82}$ | $\mathbf{65.21_{\pm3.23}}$ | $53.33_{\pm6.80}$ |

## 4.3 RESULTS

Our experiments are designed to answer three questions: (i) does explicit low-rank edge–edge modeling improve performance over strong FC baselines under a unified protocol, (ii) how does atlas choice affect absolute performance and the relative ranking of models, and (iii) how sensitive is EdgeQuad to key design choices such as ROI ordering and quadratic rank?

**Baselines.** We follow a standardized evaluation protocol to benchmark recent methods for brain networks. As general-purpose graph models, we include *GCN* (Kipf & Welling, 2017), a spectral graph convolution capturing local neighborhood structure, and *GPS* (Rampášek et al., 2022), a hybrid GNN–Transformer that combines message passing with global attention and positional encodings. To represent domain-specific classics, we evaluate *BrainNetCNN* (Kawahara et al., 2017), which introduces edge-to-edge, edge-to-node, and node-to-graph convolutions tailored for connectomes. Among recent specialized architectures, we consider three state-of-the-art approaches: *ALTER* (Yu et al., 2024), a brain graph Transformer with adaptive long-range modeling via biased random walks; *BQN* (Yang et al., 2025), a quadratic network that directly leverages adjacency structure and aligns with implicit community detection without explicit message passing; and *BioBGT* (Peng et al., 2025b), a biologically informed Transformer that encodes small-world connectivity using node-importance weighting and module-aware self-attention. We focus here on fixed-atlas supervised FC models that can be evaluated under a single unified protocol (shared seeds, splits, and capacity budget). Hierarchical pooling and multi-atlas distillation methods (Li et al., 2021; Xu et al., 2024; 2025; Tang et al., 2024a) have different ob-

Table 3: **Overall performance for ABIDE and ADNI.** Mean AUC and average AUC rank across curated atlases (AAL116, Schaefer100, Harvard–Oxford). Lower rank is better.

| | ABIDE | | ADNI | |
|---|---|---|---|---|
| Method | mean AUC | avg rank | mean AUC | avg rank |
| GCN | 64.37 | 4.67 | 80.00 | 4.33 |
| GPS | 62.37 | 5.67 | 69.20 | 6.67 |
| BrainNet | 65.68 | 4.67 | 77.70 | 5.00 |
| BioBGT | 57.04 | 7.00 | 78.90 | 4.33 |
| ALTER | 67.18 | 2.67 | 78.70 | 4.67 |
| BQN | 70.07 | 2.33 | 91.10 | 2.00 |
| **EdgeQuad** | **71.57** | **1.00** | **94.50** | **1.00** |

jectives (coarsening, contrastive learning, multi-atlas fusion) and training regimes, and are therefore treated as complementary rather than direct baselines in our setting (see Section 2.1).

**Results.** We evaluate the models on four benchmarks (see Tables 2 and 4). **ABIDE.** On the three curated/functional atlases (AAL116, Schaefer100, Harvard–Oxford), `EdgeQuad` is consistently best in AUC/ACC; BQN, the closest quadratic baseline, is typically second, with ALTER/BioBGT/GPS/BrainNet behind. On clustering atlases (Ward100, KMeans100) all methods drop and leaders shuffle (e.g., BioBGT on Ward100), indicating that noisy, non-coherent parcels degrade correlation graphs. This supports our premise: shallow, explicit *edge–edge* interactions excel when parcels are functionally coherent, and atlas quality can flip "SOTA vs. SOTA" rankings. **ADNI.** The pattern repeats: `EdgeQuad` attains the best AUC on AAL116/Schaefer100/Harvard–Oxford and stays within a few points on ACC; BQN is the closest challenger. On clustering atlases, accuracy declines for all; BioBGT leads Ward100 while `EdgeQuad` tops KMeans100. **PPMI (4-class).** This harder task compresses margins and exposes baseline instability across atlases. `EdgeQuad` is top or near-top on AAL116 and KMeans100 and yields the best ACC on Ward100, while GCN narrowly leads AUC on Harvard–Oxford. Overall, `EdgeQuad` remains competitive and stable. **ADHD200.** With Craddock-200 atlas (Yang et al., 2025), `EdgeQuad` achieves the highest AUC/ACC, outperforming BQN, ALTER, GPS, and BrainNet; BioBGT is out-of-memory at this resolution, underscoring our model's efficiency.

**Takeaway.** Across four datasets and five atlases each, `EdgeQuad` is best or a close second on functionally coherent atlases and competitive on clustering atlases. Results validate explicit low-rank quadratic edge modeling and show that *atlas choice is a consequential hyperparameter*; our standardized cross-atlas protocol is essential for fair, reproducible comparison.

**Why atlas choice matters.** *Schaefer, AAL, and Harvard/Oxford* are functionally informed or anatomically curated parcellations, whereas *Ward100* and *KMeans100* are generic clustering-based atlases (See Appendix A.1 for details). We observe markedly higher and more consistent accuracy with the for-

Table 4: **ADHD-200 results.** Mean±std over 5 seeds; best per column in **bold blue**, second-best in blue.

| Method | AUC | ACC |
|---|---|---|
| GCN | 72.56$_{\pm 5.49}$ | 64.89$_{\pm 4.75}$ |
| GPS | 75.58$_{\pm 4.28}$ | 67.11$_{\pm 5.33}$ |
| BrainNet | 73.72$_{\pm 4.90}$ | 70.40$_{\pm 6.50}$ |
| ALTER | 76.52$_{\pm 3.56}$ | 59.20$_{\pm 21.67}$ |
| BQN | 76.50$_{\pm 1.29}$ | 69.84$_{\pm 3.13}$ |
| EdgeQuad | **84.68$_{\pm 0.93}$** | **77.96$_{\pm 3.78}$** |

mer than with the latter. This is expected: functionally/anatomically defined atlases produce ROIs that are spatially contiguous and functionally coherent, yielding higher within-parcel homogeneity and more reliable ROI time series. Pearson FC is then a stable statistic, Fisher-$z$ correlations concentrate and edges show better test–retest/site reliability, so edge patterns align with known mesoscale organization (modules, hubs). In contrast, clustering atlases (e.g., unconstrained KMeans or variance-driven Ward) often mix disparate voxels or split canonical networks. The resulting ROI signals have lower SNR, larger estimation variance, and weaker modular structure, which degrades any graph model that consumes the correlation matrix. Consistent with this picture, Table **??** shows that for ABIDE and ADNI curated atlases (AAL116, Schaefer100, Harvard/Oxford) often outperform clustering-based parcellations (Ward100, KMeans100) by large margins, whereas PPMI exhibits a much smaller cross-atlas spread, indicating that the benefit of atlas curation is disease dependent.

**Implications for reproducible evaluation.** These findings show that "SOTA vs. SOTA" comparisons are confounded by atlas choice: a model that looks strong on a coherent atlas may collapse on noisy parcellations, and vice versa. *Our contribution is to make this dependence explicit and measurable* by treating the atlas as a first-class hyperparameter and *evaluating all methods under a single, standardized protocol.* This surfaces hidden failure modes and favors models that remain stable as atlas quality varies. Practically, for Pearson-based FC with graph models, one should (i) prefer functionally informed, spatially constrained atlases (e.g., Schaefer) or well-established anatomical ones (AAL/Harvard), (ii) verify parcel homogeneity and degree-normalize FC, and (iii) report cross-atlas results with fixed seeds/splits. Under these conditions, `EdgeQuad` remains accurate and stable across seeds and atlas choices, while several recent baselines exhibit high variance or failure modes, underscoring the value of our protocol as a foundation for reproducible progress. In particular, the atlas can induce performance differences that are comparable to, or larger than, the gaps between architectures evaluated on a fixed parcellation, so it must be treated as a first-order structural design choice rather than a minor tuning knob.

## 4.4 ABLATION STUDIES

Beyond aggregate performance, we next analyze the design choices that drive these results. Here we probe *why* EdgeQuad works and how sensitive it is to key choices. We test (i) robustness to **ROI ordering**, to ensure the CNN-on-correlation does not exploit index locality, and (ii) the effect of the quadratic **rank** $k$ on accuracy/efficiency. All ablations follow the same splits, seeds, and preprocessing as the main experiments.

**ROI ordering.** We tested whether our CNN-on-correlation design depends on the arbitrary ordering of ROIs in the matrix. Since atlas indices have no inherent spatial meaning, a model that exploits index locality (e.g., contiguous blocks) could overfit to ordering artifacts. Table 5 permutes ROI order in five ways (identity, random, reverse, block-swap, circular) and reports AUC/ACC across atlases. EdgeQuad is essentially invariant on functional/anatomical atlases (AAL116, Schaefer100, Harvard–Oxford): changes are within 1 AUC / 2 ACC points and often inside the seed variance, indicating the dual-ASPP + quadratic block does not rely on row/column proximity. On clustering atlases (Ward/KMeans), performance is lower overall and slightly more variable, but still shows no systematic gain from any particular ordering. This supports that our gains stem from edge–edge modeling rather than index-local patterns.

**Rank Parameter.** We analyze the impact of the *rank parameter* in the quadratic block on the ABIDE_AAL116 dataset (Figure 2). As shown, very low ranks underfit, yielding limited accuracy (60–62%) and ROC-AUC (69–70%). Increasing the rank improves expressiveness and performance, with intermediate values achieving the best trade-off: accuracy peaks at 66.2% and ROC-AUC at 72.6%. Beyond this point, higher

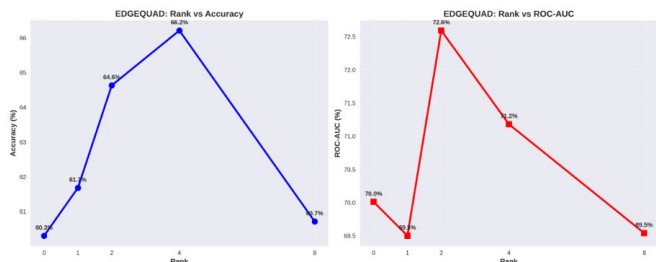

Figure 2: **Effect of the rank parameter.** accuracy (left) and ROC-AUC (right) peak at intermediate ranks, while very low or very high ranks reduce performance.

ranks begin to overfit and both metrics decline. These results highlight the central role of the rank parameter in balancing model capacity and generalization, with mid-range values consistently delivering the most stable gains for brain network classification. A complementary depth sensitivity experiment is presented in Appendix B.5.

**Ablation of architectural components.** Table 6 evaluates the contribution of key design choices on the ADHD200 dataset with the Craddock-200 atlas. The full EdgeQuad model achieves the best accuracy and AUC (77.96 and 84.68). Removing the connectivity-side ASPP branch (*w/o Edge ASPP*) leads to a clear drop of about 6–7 points in both ACC and AUC, indicating that refining $C$ with multi-scale neighborhoods is important. Eliminating the content gate (*w/o Content Gate*) produces a similar degradation, showing that data-driven fusion of linear and quadratic paths is beneficial. Replacing cluster pooling with global mean pooling causes the

Table 6: **Contribution of EdgeQuad components.** On ADHD200 (Craddock-200), each major component contributes to overall performance: removing the Edge ASPP, content gate, structured pooling, or degree-aware normalization leads to clear drops in ACC and AUC.

| Ablation | ACC | AUC |
|---|---|---|
| Full Model | **77.96**±3.78 | **84.68**±0.93 |
| w/o Edge ASPP | 71.36±6.09 | 79.73±6.59 |
| w/o Content Gate | 70.18±8.39 | 79.05±8.92 |
| Global Mean Pooling | 65.55±4.45 | 71.72±4.97 |
| RMSNorm | 71.12±5.61 | 79.10±4.52 |
| No Normalization | 72.54±6.37 | 80.74±5.62 |

Table 5: **ROI ordering ablation (ABIDE).** Mean±std AUC/ACC over 5 seeds for different ROI permutations across five atlases.

| Permutations | AAL116 | | SCHAEFER100 | | HARVARD48 | | WARD100 | | KMEANS100 | |
|---|---|---|---|---|---|---|---|---|---|---|
| | AUC | ACC | AUC | ACC | AUC | ACC | AUC | ACC | AUC | ACC |
| IDENTITY | 70.60±0.93 | 63.86±0.61 | 72.92±0.76 | 65.96±2.21 | 71.18±1.17 | 66.51±2.20 | 52.54±5.14 | 51.10±2.42 | 53.33±7.04 | 53.45±1.12 |
| RANDOM | 70.61±0.49 | 64.66±1.51 | 72.75±1.43 | 64.81±1.14 | 70.76±1.66 | 65.63±2.01 | 52.44±1.80 | 51.39±1.03 | 51.90±2.55 | 52.76±0.16 |
| REVERSE | 70.09±1.00 | 64.51±1.76 | 72.83±1.12 | 64.68±2.21 | 71.72±1.75 | 66.22±2.03 | 51.32±2.78 | 50.48±2.47 | 51.39±4.70 | 52.97±0.84 |
| BLOCK_SWAP | 70.54±0.99 | 63.26±1.57 | 72.87±0.80 | 66.07±2.36 | 71.31±1.04 | 64.23±1.56 | 51.47±2.89 | 51.68±1.57 | 50.32±5.94 | 51.79±1.79 |
| CIRCULAR | 70.51±1.25 | 62.27±1.69 | 72.91±1.27 | 66.69±2.19 | 70.36±1.94 | 64.67±2.78 | 52.05±3.30 | 50.23±2.18 | 50.27±5.50 | 52.83±1.33 |

largest decline, confirming that structured pooling over clusters is preferable to naive global aggregation. Finally, alternative or absent normalization (*RMSNorm*, *No Normalization*) consistently underperforms the full model, suggesting that the chosen normalization scheme contributes meaningfully to stability and overall performance. *Together with the $k=0$ ablation above, these results indicate that the quadratic head provides the main gain over a purely linear model, while Edge ASPP, content gating, cluster pooling, and degree-aware normalization each contribute additional improvements.*

**Hyperparameter grid.** To ensure that the unified hyperparameter grid does not disadvantage any baseline, we also ran BrainNetCNN, ALTER, BQN, and `EdgeQuad` with their recommended configurations from the original papers and report the resulting performance on ABIDE AAL116 in Table 11. These per model settings lie inside our global grid (same training budget and optimizer family), and `EdgeQuad` still achieves the highest AUC and competitive accuracy, supporting that our main conclusions do not depend on a restrictive or biased choice of hyperparameters. See Appendix B.2 for further details.

## 5 CONCLUSION

We revisited FC modeling with a lightweight *edge-image* encoder that applies dual ASPP (on features and directly on connectivity) and a low-rank quadratic branch to make edge–edge interactions explicit. Under a unified protocol, aligned seeds/splits, harmonized preprocessing, and evaluation across all common atlases, `EdgeQuad` is consistently accurate, well-calibrated, and efficient, outperforming recent graph/transformer baselines on ABIDE and ADNI while showing markedly lower variance across seeds and atlases. Our analysis clarifies why: the quadratic block implements rank-$k$ degree-2 interactions localized by dual ASPP and enjoys Lipschitz-type stability with respect to both features and refined connectivity.

Beyond the model, our standardized evaluation decouples architectural merit from protocol choices and offers a reproducible basis for future work; we release code, exact configs, and per-seed logs. Looking ahead, we will (i) extend to alternative FC estimators (partial correlation, coherence) and dynamic FC, (ii) study cross-site pretraining and domain adaptation, (iii) probe permutation sensitivity and atlas design/selection, and (iv) expand robustness tests (calibration, OOD/site shift) and clinical utility analyses.

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

# Appendix

## A  How fMRI Becomes a Brain Connectivity Network

This appendix gives an ML-friendly overview of how resting-state fMRI is converted into a subject-level brain network (a weighted graph), with technical details in (Xu et al., 2023).

**What fMRI gives you.**  An fMRI scan records a time series of activity for many small locations in the brain. Think of it as thousands of synchronized time series.

**From many time series to a graph.**  To make this data usable for ML, we summarize it as a subject-level graph.

1. **Define the nodes.** Split the brain into $N$ regions of interest (ROIs) using an atlas. Each ROI is one node.
2. **Get one signal per node.** Average the voxel signals inside each ROI to obtain one BOLD time series per node.
3. **Clean the signals.** Remove obvious noise such as motion and slow drifts and put every subject into a common space. This step follows standard neuroimaging practice; see (Xu et al., 2023) for details and choices.
4. **Compute connectivity.** Measure how similar two ROI time series are. Pearson correlation is the common default. This gives an $N \times N$ matrix of pairwise values.
5. **Build the graph.** Nodes are ROIs. Edge weights are the connectivity values. You can keep all weights as a weighted graph or keep only the strongest connections. The diagonal is ignored.

**Static vs dynamic views.**  You can compute one connectivity matrix using the full scan (static FC), or compute many matrices over short windows to capture changes over time (dynamic FC). Static FC is simpler and common for benchmarking. Dynamic FC is useful when temporal changes matter.

**What the ML model sees.**  After these steps, the ML input is:

- a connectivity matrix (often treated as an image or an adjacency for a graph model),
- optional node features such as the ROI time series or summary features derived from them.

**Good practice.**  When constructing brain networks, it is important to document the key design choices: the atlas used for defining regions, the preprocessing steps applied to the BOLD signals, the method used to estimate connectivity, and any thresholding or normalization applied to the resulting matrices. For multi-site datasets, splits should be stratified by site to avoid data leakage. A detailed end-to-end discussion of these steps, including recommended defaults and alternatives, can be found in (Xu et al., 2023).

### A.1  Brain Parcellation Methods

We summarize the atlases used in our study; for comprehensive reviews of anatomical and functional parcellations, see (Arslan et al., 2018; Eickhoff et al., 2018). See Table 8 for a summary.

**Schaefer (functional, multi-resolution).**  A gradient- and network-informed atlas that defines spatially contiguous, functionally coherent parcels aligned to canonical systems (DMN, FPN, VIS). Available at 100/200/400 resolutions, it is commonly used in benchmarks and yields homogeneous ROI signals with clear mesoscale modularity—favorable for Pearson FC and reproducible across sites.

**AAL (anatomical, curated).**  The Automated Anatomical Labeling atlas segments the brain via macrostructural landmarks and is widely used in clinical and multi-site cohorts. Although not functionally defined, its contiguity and cross-subject consistency provide stable ROI signals and reliable FC—often serving as a strong baseline in benchmarks.

Table 7: Unified training/evaluation protocol (applies to all methods unless otherwise noted).

| Aspect | Setting |
|---|---|
| Data splits | Random subject splits: 70% train / 10% val / 20% test |
| Runs / seeds | 5 runs with different seeds; results averaged |
| Model selection | Best (minimum) validation loss checkpoint per seed |
| Epochs / batch size | 200 epochs; batch size 16 |
| Optimizer | Adam; initial LR $1 \times 10^{-4}$; weight decay $1 \times 10^{-4}$ |
| LR schedule | Decay toward $1 \times 10^{-5}$ during training |
| Activations | {GELU, LeakyReLU, ELU} (chosen by validation) |
| Depth $k$ | {$1, 2, 3, 4, 5$} (chosen by validation) |
| Dropout | {$0.0, 0.1, 0.2, 0.3$} (chosen by validation) |
| Fairness | Same splits, seeds, and grids across methods when compatible |

**Harvard–Oxford (anatomical, probabilistic).** A probabilistic atlas covering cortical and subcortical regions, thresholded into discrete ROIs. It balances anatomical fidelity with practical parcel sizes, producing coherent signals and good cross-site comparability—yielding consistent Pearson FC performance in our setting.

**Ward100 (clustering, variance/min-size constrained).** Voxel clustering with Ward's linkage (typically with spatial and size constraints). While uniform parcel sizes are attractive, boundaries may cut across functional areas, reducing within-ROI homogeneity and weakening modular structure—conditions under which correlation-based graphs and downstream models can become unstable.

**KMeans100 (clustering, unconstrained).** Unconstrained $k$-means voxel clustering can produce discontiguous or mosaic parcels that mix functionally distinct tissue. The resulting ROI averages have lower SNR and higher estimation variance, which degrades Pearson correlations and often leads to brittle downstream performance across seeds and sites.

*Other commonly used atlases include* Desikan–Killiany (DKT) and Destrieux (anatomical), Gordon and Power-264 (functional nodes), Yeo-7/17 and Glasser HCP-MMP1.0 (cortex-wide hybrids), Brainnetome, Shen, and Craddock-200 (fine-grained functional), which we note for completeness.

Table 8: Parcellations and graph representation used across all datasets.

| Name | Type | #ROIs | Notes |
|---|---|---|---|
| AAL | Atlas | 116 | Anatomical parcels delineated by sulcal landmarks. |
| Harvard–Oxford | Atlas | 48 | Gyral-based cortical regions from probabilistic atlases. |
| Schaefer | Atlas | 100 | Gradient-weighted MRF functional parcels. |
| k-means | Clustering | 100 | Subject-wise voxel clustering into non-overlapping regions. |
| Ward | Clustering | 100 | Agglomerative, variance-minimizing voxel clustering. |

*Edges:* Pearson-correlation FC, fully weighted (signed).    *Node features:* ROI time series (or derived).

# B   MORE EXPERIMENTAL DETAILS

## B.1   ADDITIONAL METRICS

In Table 9,We evaluate models using four complementary measures common to machine learning and clinical practice: area under the ROC curve (AUC), accuracy (ACC), sensitivity (SEN; true positive rate), and specificity (SPE; true negative rate). **AUC** summarizes class separability across thresholds, robust to imbalance. **ACC** reports the overall proportion of correct predictions, providing a simple summary. **SEN** quantifies the fraction of positives correctly identified, critical when missed detections carry high cost, while **SPE** measures the fraction of negatives correctly rejected, guarding against false alarms. Together, these metrics capture discrimination, correctness, and error trade-offs.

Table 9: **Additional Metrics.** We report extended results for all methods under a unified protocol (aligned splits, preprocessing, and hyperparameter grids) across five atlases. For ABIDE and ADNI (binary classification), values are mean±std AUC/ACC over 5 seeds. For PPMI (4-class), we report Macro AUC and Accuracy.

**ABIDE (Binary)**

| Method | AAL116 | | | | SCHAEFER100 | | | | HARVARD48 | | | |
|---|---|---|---|---|---|---|---|---|---|---|---|---|
| | AUC | ACC | SEN | SPE | AUC | ACC | SEN | SPE | AUC | ACC | SEN | SPE |
| GCN | 64.51±2.58 | 60.39±3.43 | 61.57±6.97 | 59.22±1.92 | 64.75±3.10 | 60.98±3.79 | 65.10±2.60 | 56.86±6.68 | 63.85±2.12 | 60.39±2.29 | 63.92±5.49 | 56.86±2.77 |
| GPS | 60.93±2.05 | 58.82±2.48 | 52.55±2.88 | 65.10±3.37 | 63.37±3.55 | 58.63±3.84 | 60.39±5.46 | 56.86±4.11 | 62.80±1.43 | 58.04±1.90 | 55.29±4.71 | 60.78±3.51 |
| BrainNet | 66.80±2.12 | 62.16±2.43 | 57.25±4.00 | **67.45±3.64** | 67.55±1.34 | 64.31±1.37 | 67.84±6.75 | 61.18±5.46 | 62.68±3.33 | 58.53±2.27 | 57.25±1.92 | 61.18±4.19 |
| BioBGT | 56.55±2.93 | 55.88±2.99 | 71.70±31.30 | 38.78±31.47 | 59.28±1.74 | 53.92±2.18 | **88.68±28.71** | 26.33±7.34 | 55.28±2.29 | 51.96±1.15 | **88.11±19.48** | 22.34±9.76 |
| ALTER | 69.22±1.66 | 58.04±3.64 | **82.35±8.94** | 33.73±15.56 | 68.40±1.10 | 64.12±1.18 | 65.10±2.88 | **63.14±3.37** | 63.93±1.22 | 57.84±1.52 | 58.04±4.22 | 57.65±3.64 |
| BQN | 68.59±1.48 | 63.35±3.22 | 64.08±11.57 | 62.59±15.72 | 71.36±0.73 | 63.83±0.69 | 65.92±11.43 | 62.04±10.41 | 70.25±1.31 | 62.97±1.63 | 68.78±5.65 | 57.59±7.46 |
| EdgeQuad | **70.60±0.93** | **63.86±0.61** | 63.67±6.51 | 63.89±6.90 | **72.92±0.76** | **65.96±2.21** | 70.82±10.71 | 61.48±13.14 | **71.18±1.17** | **66.51±2.20** | 63.47±5.89 | **69.26±7.84** |

| Method | WARD100 | | | | KMEANS100 | | | |
|---|---|---|---|---|---|---|---|---|
| | AUC | ACC | SEN | SPE | AUC | ACC | SEN | SPE |
| GCN | 50.10±3.49 | 49.61±2.60 | 59.22±8.54 | 40.00±5.35 | 54.08±3.48 | **53.53±3.65** | 67.14±9.42 | 36.96±11.00 |
| GPS | 49.86±4.47 | 50.98±2.56 | 64.31±7.27 | 37.65±3.37 | 51.94±3.96 | 51.57±3.85 | 61.79±3.11 | 39.13±10.38 |
| BrainNet | 49.75±2.27 | 50.49±1.89 | 61.57±5.20 | 38.82±7.68 | 47.88±2.49 | 48.43±2.64 | 51.43±6.91 | 46.09±5.90 |
| BioBGT | **60.11±4.69** | **55.88±3.01** | **82.45±17.41** | 26.33±14.42 | 47.57±2.07 | 48.04±1.49 | **73.34±17.89** | 21.76±19.33 |
| ALTER | 50.97±1.89 | 48.43±1.00 | 50.20±1.57 | 46.67±3.14 | 48.84±0.97 | 51.57±2.29 | 57.50±3.64 | 44.35±3.79 |
| BQN | 55.87±5.07 | 52.10±3.93 | 74.29±12.59 | 31.85±16.68 | **54.30±2.85** | 50.74±2.16 | 31.84±39.96 | 67.78±39.84 |
| EdgeQuad | 52.54±5.14 | 51.10±2.42 | 31.02±38.72 | **69.26±39.16** | 53.33±7.04 | 53.45±1.12 | 8.98±11.46 | **93.33±8.35** |

**ADNI (Binary)**

| Method | AAL116 | | | | SCHAEFER100 | | | | HARVARD48 | | | |
|---|---|---|---|---|---|---|---|---|---|---|---|---|
| | AUC | ACC | SEN | SPE | AUC | ACC | SEN | SPE | AUC | ACC | SEN | SPE |
| GCN | 88.50±6.63 | 81.54±6.15 | 77.50±5.00 | 88.00±9.80 | 80.50±6.96 | 69.23±6.88 | 65.00±9.35 | 76.00±8.00 | 71.00±5.83 | 66.15±6.15 | 52.50±9.35 | 88.00±9.80 |
| GPS | 83.50±4.06 | 73.85±3.77 | 75.00±6.12 | 72.00±9.80 | 67.00±3.67 | 61.54±6.88 | 67.50±6.12 | 52.00±9.80 | 57.00±3.32 | 49.23±3.77 | 42.50±6.12 | 60.00±0.00 |
| BrainNet | 85.50±3.32 | 76.15±3.77 | 80.00±6.12 | 68.00±9.80 | 74.00±2.55 | 73.08±3.44 | 85.00±5.00 | 56.00±8.00 | 73.50±6.63 | 66.92±6.71 | 62.50±7.91 | 80.00±17.89 |
| BioBGT | 71.43±15.76 | 61.46±14.27 | 50.00±34.96 | 65.71±36.81 | 81.90±8.38 | 61.54±4.87 | 90.00±5.00 | 65.71±23.21 | 83.33±8.42 | 76.92±8.97 | 66.67±17.89 | 85.71±16.65 |
| ALTER | 85.00±1.58 | 81.54±3.77 | 85.00±5.00 | 76.00±8.00 | 75.00±1.58 | 73.85±3.77 | 82.50±6.12 | 60.00±0.00 | 76.00±2.55 | 70.77±3.08 | 65.00±5.00 | 80.00±0.00 |
| BQN | 93.53±1.00 | 80.69±2.76 | 90.00±3.33 | 74.12±2.88 | 94.12±2.03 | 78.62±2.58 | 96.67±6.67 | 65.88±6.86 | 85.69±3.91 | 75.17±7.98 | 90.00±6.24 | 64.71±15.78 |
| EdgeQuad | 96.76±0.73 | 80.96±6.42 | 95.00±6.67 | 71.76±14.60 | 95.39±1.85 | 76.11±2.97 | 98.33±3.33 | 60.00±5.76 | 91.37±4.21 | 65.91±1.21 | 98.33±3.33 | 43.53±2.88 |

| Method | WARD100 | | | | KMEANS100 | | | |
|---|---|---|---|---|---|---|---|---|
| | AUC | ACC | SEN | SPE | AUC | ACC | SEN | SPE |
| GCN | 52.50±5.24 | 30.00±12.75 | 76.00±8.00 | 47.69±7.54 | 55.50±10.30 | 50.77±3.77 | 30.00±6.12 | 84.00±14.97 |
| GPS | 54.12±11.87 | 40.00±4.21 | 84.31±3.79 | 14.00±3.86 | 52.50±12.04 | 43.08±10.43 | 30.00±12.75 | 64.00±14.97 |
| BrainNet | 56.50±14.88 | 52.31±8.28 | 50.00±25.00 | 56.00±32.00 | 57.50±11.51 | 50.77±14.47 | 42.50±26.93 | 68.00±32.50 |
| BioBGT | 71.43±12.87 | 58.46±12.50 | 63.33±22.11 | 54.29±35.46 | 57.62±9.57 | 52.31±12.31 | 23.33±17.00 | 77.14±26.50 |
| ALTER | 62.50±8.22 | 49.23±10.43 | 32.50±15.00 | 76.00±8.00 | 69.00±6.63 | 47.69±7.54 | 15.00±12.25 | 100.00±0.00 |
| BQN | 60.10±8.18 | 55.17±6.90 | 43.33±37.79 | 63.53±36.34 | 68.33±5.38 | 59.28±4.57 | 36.67±13.54 | 83.53±10.12 |
| EdgeQuad | 58.76±8.35 | 56.73±6.74 | 51.67±16.16 | 60.00±15.96 | 69.71±13.49 | 61.01±1.72 | 33.30±4.08 | 100.00±0.00 |

**PPMI (4-Class)**

| Method | AAL116 | | | | SCHAEFER100 | | | | HARVARD48 | | | |
|---|---|---|---|---|---|---|---|---|---|---|---|---|
| | AUC | ACC | SEN | SPE | AUC | ACC | SEN | SPE | AUC | ACC | SEN | SPE |
| GCN | 60.66±6.47 | 57.89±3.33 | **82.50±1.27** | 24.14±2.47 | **67.11±4.03** | 51.58±6.14 | 80.28±4.76 | 25.71±2.01 | 59.91±4.67 | **58.95±3.94** | 88.33±3.24 | 22.07±2.72 |
| GPS | 60.93±2.05 | **58.82±2.48** | 52.55±2.88 | 65.10±3.37 | 65.88±2.42 | 40.00±5.37 | 75.28±3.19 | 29.14±12.69 | 57.82±3.26 | 45.26±4.21 | 75.14±2.03 | 27.86±11.43 |
| BrainNet | **62.44±8.62** | 53.68±6.98 | 79.31±3.81 | 27.29±4.73 | 62.32±4.18 | 50.00±3.72 | 80.14±2.69 | 30.00±13.92 | 53.27±3.10 | 52.63±6.86 | 82.36±2.97 | 24.86±5.26 |
| BioBGT | 47.26±16.48 | 57.89±0.00 | 25.00±0.00 | **75.62±1.25** | 52.13±5.14 | **57.89±0.00** | 24.09±1.11 | 75.00±0.00 | 45.57±5.70 | **57.89±0.00** | 25.00±0.00 | **75.00±0.00** |
| ALTER | 61.97±5.48 | 52.63±2.43 | 78.70±1.31 | 24.17±1.18 | 61.56±1.51 | 51.58±6.14 | **83.06±1.68** | 22.79±3.86 | 55.79±7.26 | 61.05±5.37 | 85.69±2.62 | 31.57±4.29 |
| BQN | 50.00±0.00 | 31.67±11.67 | 4.47±2.97 | **95.16±2.62** | 50.00±0.00 | 33.33±10.54 | 3.30±2.09 | **96.80±2.13** | 50.00±0.00 | 22.92±3.23 | 0.00±0.00 | **100.00±0.00** |
| EdgeQuad | **66.34±2.17** | 49.17±1.67 | 34.10±10.75 | 80.82±3.39 | 59.63±4.05 | 47.50±5.00 | 24.36±7.00 | 77.70±5.01 | **58.69±5.12** | 53.33±7.52 | 28.39±3.46 | 81.46±3.91 |

| Method | WARD100 | | | | KMEANS100 | | | |
|---|---|---|---|---|---|---|---|---|
| | AUC | ACC | SEN | SPE | AUC | ACC | SEN | SPE |
| GCN | **55.62±4.60** | **60.00±6.32** | **83.06±3.61** | 28.36±3.29 | 54.63±9.88 | 51.58±2.11 | **87.22±2.83** | 12.50±4.47 |
| GPS | 54.12±11.87 | 40.00±4.21 | 84.31±3.79 | 14.00±3.86 | 53.69±12.36 | 50.53±12.72 | 85.14±6.17 | 20.00±6.03 |
| BrainNet | 45.58±10.79 | 46.32±4.88 | 75.56±3.39 | 20.79±5.26 | 30.78±8.36 | 39.47±6.00 | 70.00±3.50 | 16.93±3.20 |
| BioBGT | 55.42±12.35 | 57.89±0.00 | 25.00±0.00 | 75.00±0.00 | 48.27±7.42 | **57.89±0.00** | 25.00±0.00 | 75.00±0.00 |
| ALTER | 41.45±7.60 | 55.79±9.76 | 81.39±3.21 | 25.86±3.44 | 57.60±1.89 | 54.74±2.58 | 77.36±1.21 | 23.93±3.41 |
| BQN | 50.00±0.00 | 28.33±26.21 | 0.00±0.00 | **100.00±0.00** | 50.00±0.00 | 18.33±21.06 | 0.00±0.00 | **100.00±0.00** |
| EdgeQuad | 45.04±4.32 | 55.83±4.82 | 18.12±9.23 | 81.49±9.58 | **65.21±3.23** | 53.33±6.80 | 16.01±4.23 | 88.86±4.55 |

In Table 10, we present AUC/ACC for the ADHD-200 dataset with the Craddock-200 parcellation under our unified protocol; SEN/SPE are reported in extended tables. The BioBGT model is omitted due to out-of-memory (OOM).

Table 10: **ADHD-200 results.** Mean±std over 5 seeds; best per column in **bold blue**, second-best in blue.

| Method | AUC | ACC | SEN | SPE |
|---|---|---|---|---|
| GCN | 72.56±5.49 | 64.89±4.75 | **78.40±5.99** | 48.00±8.12 |
| GPS | 75.58±4.28 | 67.11±5.33 | 75.20±11.70 | 57.00±6.78 |
| BrainNet | 73.72±4.90 | 70.40±6.50 | 58.00±4.00 | 64.89±2.69 |
| ALTER | 76.52±3.56 | 59.20±21.67 | 70.00±30.33 | 64.00±7.08 |
| BQN | 76.50±1.29 | 69.84±3.13 | 61.74±13.01 | 75.32±11.62 |
| EdgeQuad | **84.68±0.93** | **77.96±3.78** | 73.48±10.42 | **82.13±4.78** |

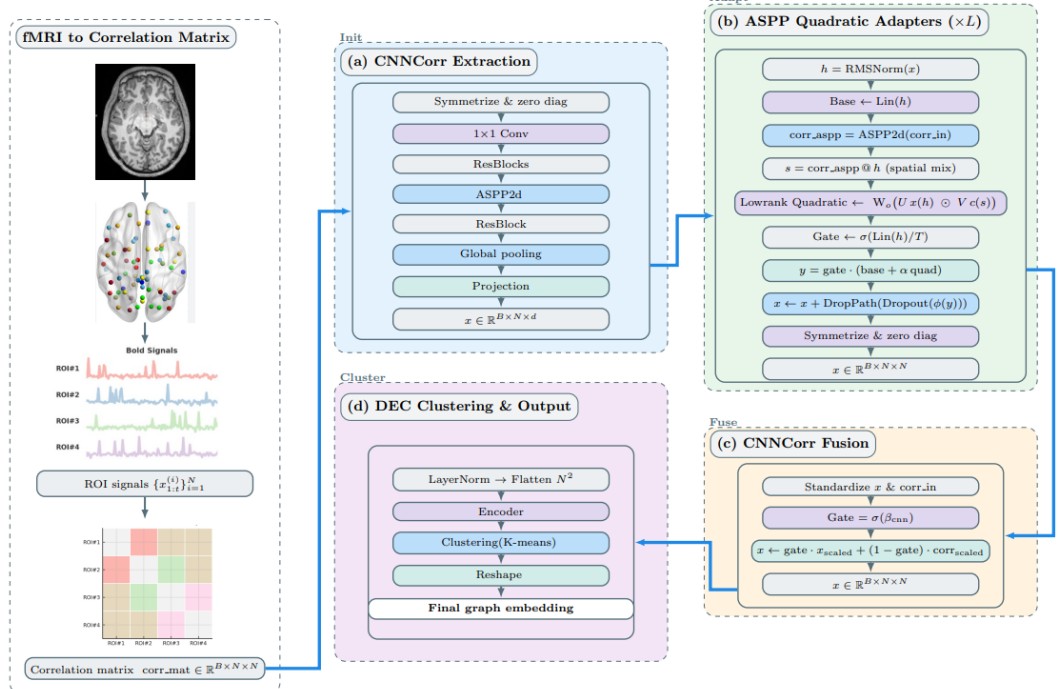

Figure 3: **EdgeQuad Detailed Flowchart** Resting-state fMRI is parcellated into ROI BOLD signals and converted to a Pearson correlation matrix. A CNN with dual ASPP extracts features and applies low-rank quadratic adapters to make edge–edge effects explicit, then a global fusion and clustering readout produce the final graph embedding for prediction.

## B.2 UNIFIED HYPERPARAMETER GRID AND PER-MODEL CONFIGURATIONS

In the main experiments we adopt a unified hyperparameter grid across all baselines and atlases. This follows common practice in recent work on graph transformers and brain network models, where a shared budget helps avoid cases in which certain methods benefit from a larger or more favorable tuning schedule. The ranges in our grid were chosen to include the values reported as effective in closely related studies such as BQN (*Do We Really Need Message Passing?*) and ALTER (*Long-Range Brain Graph Transformer*). For example, both works train for 200 epochs with batch size 16, use Adam with learning rate $10^{-4}$ and weight decay $10^{-4}$, and consider depths between two and five layers with dropout between 0.0 and 0.3. Our grid explicitly covers these settings, so baselines are not restricted relative to their published configurations.

To further clarify this point, Table 11 reports, for ABIDE AAL116, the performance of each model when run with its recommended or standard hyperparameters from the corresponding papers, together with the configuration we used (epochs, batch size, learning rates, weight decay, depth, activation, and dropout). As shown, EdgeQuad remains competitive and achieves the best AUC and balanced accuracy under these per-model configurations, which is consistent with the results obtained under the unified grid in the main text.

Table 11: Per-model configurations and performance on ABIDE (AAL116) using recommended hyperparameters from the corresponding papers. Mean ± std over 5 seeds.

| Method | AUC | ACC | SEN | SPE | Epochs | Batch | Base LR | Adapter LR | WD | Layers | Activation | Dropout |
|---|---|---|---|---|---|---|---|---|---|---|---|---|
| BrainNetCNN | 66.80±1.22 | 62.16±2.43 | 57.25±4.00 | **67.45**±3.64 | 200 | 16 | 0.001 | 0.0001 | 0.0001 | 8 | ReLU | 0.1 |
| ALTER | 68.53±1.19 | 64.90±0.96 | 58.82±2.28 | 70.98±3.87 | 200 | 16 | 0.0001 | 0.0001 | 0.0001 | 2 | ReLU | 0.1 |
| BQN | 68.59±1.48 | 63.35±3.22 | 64.08±11.57 | 62.59±15.72 | 200 | 16 | 0.0001 | 0.0001 | 0.0001 | 2 | LeakyReLU | 0.5 |
| **EdgeQuad** | **70.60**±0.93 | **63.86**±0.61 | **63.67**±6.51 | **63.89**±6.90 | 200 | 16 | 0.0001 | 0.0006 | 0.001 | 3 | LeakyReLU | 0.2 |

### B.3 Computational efficiency

To complement the algorithmic complexity discussion in the main text, we report empirical compute measurements on the ABIDE dataset with the AAL116 parcellation. We compare `EdgeQuad` against all baselines under identical experimental conditions and a unified training protocol.

All experiments in Table 12 were run on a single NVIDIA H100 NVL GPU ($\approx$ 95 GB HBM3, Hopper architecture) on our HPC cluster, using the same data splits, batch size, and number of epochs as in the main experiments. For each method, we report test AUC (mean±std over 5 seeds), training throughput (graphs per second), training time per epoch, peak GPU memory, parameter count, and an estimate of the FLOPs of the prediction head (including the quadratic branch where applicable).

Table 12: **Compute profile on ABIDE (AAL116).** Mean±std over 5 seeds. Throughput and training time are measured under identical settings on a single H100 NVL GPU.

| Method | AUC | Throughput | Train time / epoch | Peak Mem | # Params | Head FLOPs |
|---|---|---|---|---|---|---|
| GCN | $64.51_{\pm2.58}$ | $1227.86_{\pm16.89}$ | $0.78_{\pm0.0064}$ | $0.3641_{\pm0.0000}$ | 32 866 | 0.0247 |
| GPS | $60.93_{\pm2.05}$ | $363.80_{\pm1.17}$ | $1.97_{\pm0.0056}$ | $0.5227_{\pm0.0000}$ | 119 970 | 0.0052 |
| BrainNet | $66.80_{\pm2.12}$ | $1256.89_{\pm27.18}$ | $0.57_{\pm0.0128}$ | $0.4344_{\pm0.0000}$ | 556 957 | 0.0733 |
| ALTER | $68.53_{\pm1.19}$ | $329.12_{\pm1.39}$ | $2.18_{\pm0.0092}$ | $0.1848_{\pm0.0000}$ | 2 189 866 | 0.5432 |
| BQN | $68.59_{\pm1.48}$ | $2042.33_{\pm32.72}$ | $0.36_{\pm0.0037}$ | $0.0672_{\pm0.0000}$ | 3 638 902 | 0.0560 |
| **EdgeQuad** | $\mathbf{70.60}_{\pm\mathbf{1.01}}$ | $1549.81_{\pm3.31}$ | $0.46_{\pm0.0054}$ | $0.1364_{\pm0.0000}$ | 1 451 750 | 0.0400 |

We note that BioBGT is excluded from this table because, on our setup, it is substantially more expensive in both memory and runtime; we plan to include a full computational profile for BioBGT in the camera-ready version. As Table 12 shows, `EdgeQuad` attains the highest AUC on ABIDE AAL116 while maintaining an efficient compute profile: its throughput and per-epoch training time are competitive with lightweight architectures such as BQN and BrainNet, and it uses fewer parameters than some transformer-based baselines (e.g., ALTER) while avoiding their higher runtime costs. These measurements empirically support our claim that explicit low-rank quadratic edge modeling can be implemented in a computationally efficient way.

### B.4 Unified Evaluation Protocol and Comparison with Prior Work

Our experiments follow the unified protocol described in Section 4.1, which fixes all dataset- and model-level choices that typically vary across studies. Concretely, for each dataset we use a **single set of stratified subject splits** (70%/10%/20% train/validation/test), evaluated over **five independent random seeds** with identical splits for all models. We harmonize preprocessing, atlas construction, and hyperparameter grids, and we adopt clear, task-appropriate model-selection rules (macro AUC for the imbalanced multi-class PPMI, minimum validation loss for ABIDE and ADHD). All reported numbers are mean ± standard deviation over the five seeds, computed from the checkpoint selected by the unified rule. This yields a single, transparent evaluation pipeline that we apply uniformly to EdgeQuad and all baselines.

To clarify how this differs from typical practice in the literature, we summarize below the evaluation settings used in several influential rs-fMRI works. In each case, the original protocol is internally consistent and reproducible within its own framework, but differs from others in ways that hinder cross-paper comparison.

**Prior evaluation protocols.**

- **BQN** (Yang et al., 2025). Evaluated on ABIDE and related cohorts under a single 7:1:2 train/validation/test split with one random seed. Model selection is based on the minimum validation loss checkpoint, and performance is reported as a single AUC/accuracy number per dataset. While this is consistent within their pipeline, the combination of (i) a single split, (ii) a single seed, and (iii) a loss-based selection rule makes the reported numbers sensitive to both random initialization and the particular choice of validation objective, and they are not directly comparable to studies that use different splits or selection criteria.

- **BioBGT** (Peng et al., 2025a). Uses an 8:1:1 subject split and selects the checkpoint with the maximum validation score (typically validation AUC or accuracy). This maximization-based rule differs from loss-based selection and is more prone to overfitting the validation set when only a single split and seed are used. Moreover, different papers choose different validation metrics, so even when the same dataset is used, the reported "best" checkpoint may be chosen under incompatible criteria.

- **ALTER** (Yu et al., 2024). Reports results on a 7:1:2 split, but typically uses the final training epoch as the evaluation checkpoint, without early stopping. This can be disadvantageous when models overfit late in training and is not directly comparable to methods that select the best epoch by validation performance. In addition, ALTER is usually evaluated with a single random split and seed, so variance across seeds and splits is not quantified.

- **BrainNetCNN** (Kawahara et al., 2017). One of the earliest and most widely used FC baselines, BrainNetCNN is evaluated on structural connectomes of preterm infants using 3-fold cross-validation, with 56 scans per fold and the constraint that all scans from a given subject lie in the same fold. Splits and seeds are rarely fixed across follow-up works, and model-selection rules vary (final epoch vs. best validation epoch), which makes it difficult to distinguish true architectural gains from differences in training and evaluation practice.

- **Generic GNN/Transformer baselines (GCN, GPS, Brain Network Transformer, etc.)**. These models are typically introduced and tuned on non-neuroimaging benchmarks (citation graphs, molecular graphs, etc.), then adapted to rs-fMRI with study-specific choices of atlas, splits, and selection criteria. In the brain-network literature, they are usually run with single seeds and unpublished split indices, and the primary metric (accuracy vs. AUC vs. macro scores) and stopping rule differ from paper to paper.

Taken together, these heterogeneous practices mean that reported numbers across prior works are often not directly comparable, even when they use the same dataset and atlas. Differences in atlas choice, split ratios, seed handling, model-selection rules, and reported metrics can easily induce several points of variation in AUC or accuracy, which is comparable to or larger than the claimed improvements of many new architectures. Our unified protocol is designed to remove this source of ambiguity: by fixing splits and seeds, harmonizing preprocessing and hyperparameter budgets, and adopting explicit, task-appropriate selection criteria, we provide a strict, reproducible testbed on which architectural contributions can be compared fairly and future methods can be evaluated without changing the experimental protocol.

### B.5 DEPTH SENSITIVITY FOR THE LOW RANK QUADRATIC INTERACTION

In addition to the low rank ablation study in the main paper, we performed a depth sensitivity experiment to provide more direct empirical evidence for the claim that the proposed low rank quadratic interaction reduces the need for deep message passing. We used the ADHD–200 dataset with a fixed random seed (42). For the baselines BrainNetCNN and GPS, we varied the network depth over $\{2, 4, 8\}$ layers. For EdgeQuad, which is intentionally shallow, we kept the depth fixed but varied the low rank parameter $k \in \{2, 4, 8\}$. For each configuration, we report macro AUC and accuracy (in percent).

Table 13 summarizes the results. Each column block corresponds to either the depth of the baseline models (2, 4, or 8 layers) or the matching low rank parameter $k$ of EdgeQuad.

Table 13: **Depth sensitivity on ADHD–200.** Baselines (BrainNetCNN, GPS) vary the number of message passing layers, while EdgeQuad varies the low rank parameter $k$ with a shallow architecture. We report macro AUC and accuracy (mean $\pm$ standard deviation) for a fixed seed.

| Model | 2 layers / $k = 2$ | | 4 layers / $k = 4$ | | 8 layers / $k = 8$ | |
|---|---|---|---|---|---|---|
| | **AUC** | **Acc** | **AUC** | **Acc** | **AUC** | **Acc** |
| BrainNetCNN | $66.34_{\pm 0.84}$ | $64.22_{\pm 3.54}$ | $66.54_{\pm 0.84}$ | $64.22_{\pm 3.54}$ | $66.34_{\pm 0.84}$ | $64.22_{\pm 3.54}$ |
| GPS | $63.28_{\pm 2.40}$ | $58.22_{\pm 2.95}$ | $64.51_{\pm 2.37}$ | $61.33_{\pm 2.27}$ | $64.01_{\pm 2.24}$ | $57.78_{\pm 3.14}$ |
| EdgeQuad | $77.23_{\pm 2.32}$ | $69.29_{\pm 0.03}$ | $84.68_{\pm 9.30}$ | $77.96_{\pm 0.04}$ | $81.16_{\pm 3.36}$ | $71.72_{\pm 0.02}$ |

We observe that BrainNetCNN and GPS either saturate or degrade as depth increases, which is consistent with depth related issues such as over squashing and over fitting on this dataset. In contrast, shallow EdgeQuad with $k > 0$ matches or exceeds the performance of the deepest baselines across these metrics, despite using far fewer message passing layers. Together with the main low rank ablation, this depth sensitivity study provides additional empirical support for our claim that the low rank quadratic interaction alleviates the need for deep message passing.

## C    INTERPRETATION OF EDGEQUAD

To complement the quantitative results, we provide a brief qualitative analysis of what `EdgeQuad` has learned on ABIDE with the AAL116 atlas. Unless otherwise noted, all visualizations are computed from the same trained model (best validation seed) and averaged over test subjects.

**Edge and ROI saliency (Fig. 4).**    This figure highlights which connections and regions most influence `EdgeQuad`'s autism vs. control predictions. On the left, we show a $116 \times 116$ *saliency matrix*, where each pixel corresponds to a pair of ROIs, i.e., an edge in the refined connectivity matrix $C'$. The color at position $(i, j)$ is the Integrated Gradients importance score for that edge: warmer colors indicate edges where small changes would most affect the model's output, while cooler colors indicate edges with minimal influence.

To obtain a region-level view, the right panel aggregates edge saliency into *ROI saliency*. For each ROI $i$, we sum the importance scores of all edges connected to it, $\mathrm{ROI}_{saliency}(i) = \sum_j S_{ij}$, where $S$ is the saliency matrix from the left panel. We then visualize the saliency of all 116 ROIs, where higher scores indicate greater importance for the model's prediction. This shows that, although the model processes the full FC matrix, its predictions rely disproportionately on a compact subset of edges and regions, providing an interpretable summary of which brain areas matter most for the classification task.

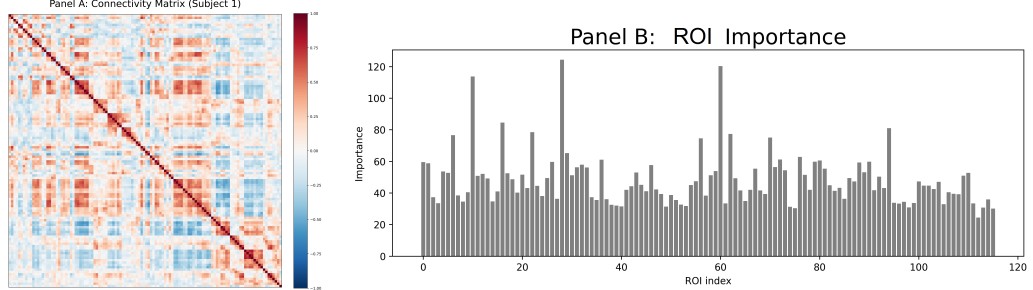

Figure 4: **Edge and ROI saliency on ABIDE (AAL116).** Left: Integrated Gradients saliency of the autism logit with respect to the refined connectivity $C'$ (averaged over test subjects and one seed). Right: corresponding ROI saliency obtained by summing incident edge saliencies, showing 116 ROIs' saliency scores.

**Quadratic rank-$k$ structure (Fig. 5).**    Figure 5 explains what the low-rank quadratic block is learning. The top row shows three individual *rank-1 components* of the quadratic head. Each $116 \times 116$ heatmap can be read as a simple pattern of *edge–edge influence*: the entry at $(i, j)$ tells us how much the quadratic block lets changes in the connection around ROI $i$ interact with connections around ROI $j$ for that component. Brighter patches indicate groups of edges that tend to co-vary in the model's decision under that single component.

The bottom panel shows the *gated sum* of all $k$ rank-1 components, i.e., the combined edge–edge influence map actually used at inference. Although the quadratic block is defined over all edges, this visualization shows that its effective influence is concentrated in a few structured mesoscale motifs rather than spread uniformly across the matrix.

**B) Rank-k Decomposition**

Rank 1    Rank 2    Rank 3

**Gated Sum (Edge-Edge Influence)**

Figure 5: **Quadratic rank-$k$ decomposition on ABIDE (AAL116).** Top: edge–edge influence heatmaps for the first three rank-1 components of the low-rank quadratic branch, showing distinct mesoscale patterns captured by each factor. Bottom: gated sum of all rank components, which represents the overall edge–edge influence used by `EdgeQuad` at inference.

**Motif-level view (Fig. 6).** Finally, we project the strongest edge–edge interactions back onto a brain graph to visualize them as concrete connection patterns. Each node on the circle is an ROI, colored by its Yeo-7 functional network (visual, somatomotor, attention, limbic, frontoparietal, default mode), and the gray lines show edges that belong to the most influential motifs according to the quadratic block. In other words, we select the connections that participate most strongly in high-scoring edge–edge interactions and plot only those, rather than the full connectome.

Many of the highlighted motifs lie *within* a single large-scale network (e.g., visual–visual or default-mode–default-mode interactions), while a smaller number of edges form *bridges* between distinct networks. This pattern indicates that EdgeQuad's quadratic terms are not acting as arbitrary second-order weights: they concentrate on structured, interpretable network-level interactions inside canonical systems, together with a few cross-network links that appear most informative for the autism vs. control classification.

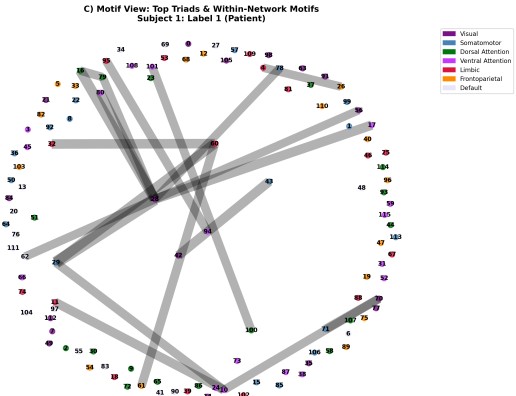

Figure 6: **Motif view of salient triads on ABIDE (AAL116).** Nodes are arranged on a circle and colored by Yeo-7 network (visual, somatomotor, dorsal attention, ventral attention, limbic, frontoparietal, default). Gray edges depict the most salient triads selected from the quadratic edge–edge influence maps. Many highlighted motifs fall within or between canonical large-scale networks, indicating that the quadratic branch concentrates on interpretable network-level interactions.

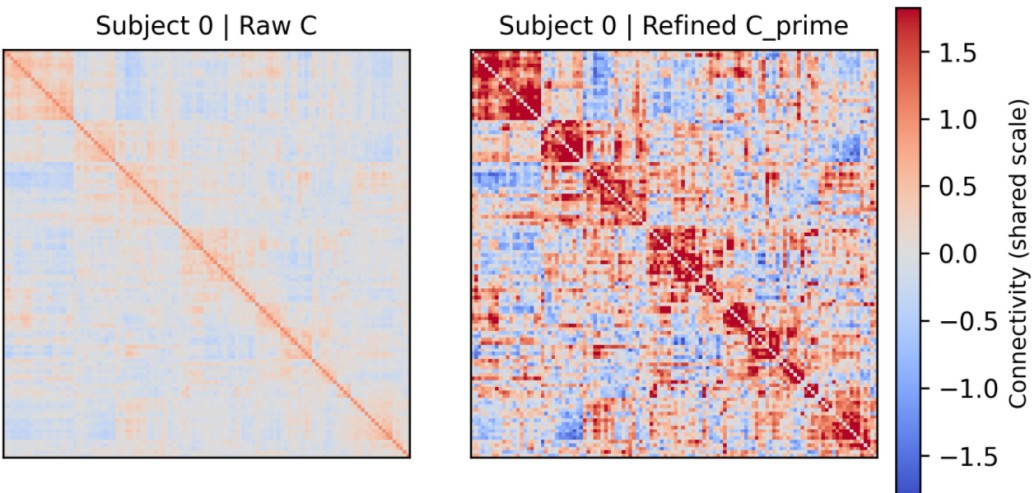

Figure 7: **Effect of dual ASPP on functional connectivity.** Left: raw Pearson correlation matrix $C$ (Fisher-$z$ transformed). Right: refined connectivity $C'$ after passing $C$ through the dual ASPP edge-image encoder. Both panels share the same color scale. The refinement sharpens block-diagonal structure and accentuates modular patterns while preserving the overall signed connectivity, illustrating how the model enhances mesoscale organization before quadratic edge–edge modeling.

## D  PROOFS OF PROPOSITIONS

**Notation recap.** $C \in \mathbb{R}^{N \times N}$ is the functional connectivity (edge image). The feature ASPP and pooling yield $X \in \mathbb{R}^{N \times d'}$. The connectivity ASPP gives $C' = \mathcal{A}_{\text{edge}}(C)$ (optionally degree-normalized $\hat{C}' = D^{-1/2} C' D^{-1/2}$). Define

$$A(X) = XV^\top \in \mathbb{R}^{N \times k}, \qquad B(X, C') = (C'X)U^\top \in \mathbb{R}^{N \times k}, \qquad Q = \big(A(X) \odot B(X, C')\big) W_o \in \mathbb{R}^{N \times d'},$$

with $U, V, W_o \in \mathbb{R}^{k \times d'}$, $k \ll d'$, and ($\odot$) the Hadamard product. Let $v_p, u_p \in \mathbb{R}^{d'}$ denote the $p$-th rows of $V, U$.

PROPOSITION 1 (FACTORIZED DEGREE-2 LIFTING)

*Statement (restated).* For fixed $U, V, W_o$, each output coordinate $Q_i(t)$ can be written as

$$Q_i(t) = \sum_{p=1}^{k} \big\langle \tilde{v}_{p,t}, x_i \big\rangle \cdot \Big\langle \tilde{u}_{p,t}, \sum_{j=1}^{N} C'_{ij} x_j \Big\rangle,$$

for suitable $\tilde{u}_{p,t}, \tilde{v}_{p,t} \in \mathbb{R}^{d'}$. Thus $Q$ realizes a rank-$k$ degree-2 polynomial in node features with coefficients *linear* in the ASPP-refined connectivity $C'$.

Intuitively, Prop. 1 shows that the block computes a sum of $k$ *separable bilinear* terms between a node's embedding $x_i$ and its ASPP-refined neighborhood summary $\sum_j C'_{ij} x_j$. Equivalently, each output coordinate is a rank-$k$ degree-2 form whose coefficients depend *linearly* on $C'$; because $C'$ arises from $\mathcal{A}_{\text{edge}}$, these coefficients are localized to the dual-ASPP receptive fields (multi-scale but spatially bounded). In FC graphs where raw node features are scarce, the encoder first builds meaningful $x_i$ from the edge image, and the quadratic block then makes edge–edge interactions *explicit* without deep propagation. This provides a shallow, parameter-efficient alternative to capturing similar effects implicitly via many layers of message passing.

**Proof of Proposition 1.**     For node $i$ and output coordinate $t$,

$$Q_i(t) = \sum_{p=1}^{k} A_{ip}\, B_{ip}\, W_o(p,t) = \sum_{p=1}^{k} \langle v_p, x_i \rangle \left\langle u_p, \sum_{j=1}^{N} C'_{ij} x_j \right\rangle W_o(p,t).$$

Set $\tilde{v}_{p,t} = W_o(p,t)\, v_p$ and $\tilde{u}_{p,t} = u_p$ to obtain

$$Q_i(t) = \sum_{p=1}^{k} \langle \tilde{v}_{p,t}, x_i \rangle \cdot \left\langle \tilde{u}_{p,t}, \sum_j C'_{ij} x_j \right\rangle,$$

a sum of $k$ separable bilinear terms. Each $Q_i(t)$ is therefore a rank-$k$ degree-2 polynomial in node features, with coefficients linear in $C'$. □

PROPOSITION 2 (FROM EDGE IMAGE TO DEGREE-2 POLYNOMIALS IN $C$)

*Statement (restated).* If the encoder from $C$ to $X$ is locally (piecewise) linear, then each $x_i$ is a linear functional of local patches of $C$. Consequently, each $Q_i(t)$ in Prop. 1 is a degree-2 polynomial in entries of $C$ supported on the union of the receptive fields of $\mathcal{A}_{\text{feat}}$ and $\mathcal{A}_{\text{edge}}$.

**Proof of Proposition 2.**     Assume the encoder from $C$ to $X$ is locally (piecewise) linear. Fix a linear region; then there exist linear maps $L_i : \mathbb{R}^{N^2} \to \mathbb{R}^{d'}$ localized to the receptive field of $\mathcal{A}_{\text{feat}} \circ \Phi_\theta$ such that $x_i = L_i(\text{vec}(C))$. Also $C'$ is linear in $C$ within the receptive field of $\mathcal{A}_{\text{edge}}$, hence $S_i = (C'X)_i = \sum_j C'_{ij} x_j$ is linear in $\text{vec}(C)$. By Prop. 1, $Q_i(t)$ is a sum of products of two linear functionals in $\text{vec}(C)$, i.e., a degree-2 polynomial supported on the union of the two receptive fields. □

PROPOSITION 3 (EXPRESSIVITY–EFFICIENCY FOR CROSS-BLOCK RANK-$k$ QUADRATICS)

*Statement (restated).* Let $z_i = [\, x_i;\ (C'X)_i \,] \in \mathbb{R}^{2d'}$. The class realized by equation **??** equals the set of *cross-block* quadratic forms

$$Q_i(t) = z_i^\top M_t z_i, \qquad M_t = \begin{bmatrix} 0 & R_t \\ 0 & 0 \end{bmatrix},$$

whose off-diagonal block $R_t \in \mathbb{R}^{d' \times d'}$ has $\text{rank}(R_t) \leq k$. As $k$ increases, this class monotonically approaches dense cross-block quadratics, using $\mathcal{O}(kd')$ parameters.

**Proof of Proposition 3.**     Write $Q_i(t) = \sum_{p=1}^{k} W_o(p,t)\langle v_p, x_i \rangle \langle u_p, (C'X)_i \rangle$. Let $a_p = [v_p; 0]$ and $b_p = [0; u_p]$. Then

$$Q_i(t) = \sum_{p=1}^{k} W_o(p,t)\, (a_p^\top z_i)(b_p^\top z_i) = z_i^\top \Big( \sum_{p=1}^{k} W_o(p,t)\, a_p b_p^\top \Big) z_i.$$

The quadratic matrix has block form $\begin{bmatrix} 0 & R_t \\ 0 & 0 \end{bmatrix}$ with $R_t = \sum_{p=1}^{k} W_o(p,t)\, v_p u_p^\top$, hence $\text{rank}(R_t) \leq k$. Conversely, any cross-block $R_t$ of rank $\leq k$ admits a decomposition $R_t = \sum_{p=1}^{k} \gamma_p v_p u_p^\top$ realized by choosing $W_o(p,t) = \gamma_p$ and setting rows of $V, U$ to $v_p^\top, u_p^\top$. □

PROPOSITION 4 (LIPSCHITZ STABILITY)

*Statement (restated).*     The class realized by equation **??** equals the set of quadratic forms in $[x_i; \sum_j C'_{ij} x_j]$ whose quadratic matrix has rank at most $k$. As $k$ increases, this class monotonically approaches the dense quadratic while using $\mathcal{O}(kd')$ parameters.

**Proof of Proposition 4.** Write $Q(X, C') = (A \odot B)W_o$ with $A = XV^\top$ and $B = (C'X)U^\top$. For perturbations $(\Delta X, \Delta C')$ and dropping second-order terms, set

$$\Delta A = \Delta X \, V^\top, \qquad \Delta B = (C'\Delta X + \Delta C'X)U^\top.$$

Then

$$Q(X+\Delta X, C'+\Delta C') - Q(X, C') = (\Delta A \odot B)W_o + (A \odot \Delta B)W_o.$$

Using $\|PQ\|_F \le \|P\|_F\|Q\|_2$ and $\|A \odot B\|_F \le \|A\|_F\|B\|_F$, we bound

$$\|A\|_F \le \|X\|_F\|V\|_2, \quad \|B\|_F \le \|C'\|_2\|X\|_F\|U\|_2, \quad \|\Delta A\|_F \le \|\Delta X\|_F\|V\|_2,$$

$$\|\Delta B\|_F \le (\|C'\|_2\|\Delta X\|_F + \|\Delta C'\|_2\|X\|_F)\|U\|_2.$$

Hence

$$\|Q(X+\Delta X, C'+\Delta C') - Q(X, C')\|_F \le \|W_o\|_2\big(\|\Delta A\|_F\|B\|_F + \|A\|_F\|\Delta B\|_F\big)$$

$$\le \|W_o\|_2\|U\|_2\|V\|_2\big(2\|C'\|_2\|X\|_F\|\Delta X\|_F + \|X\|_F^2\|\Delta C'\|_2\big).$$

Let $M_U = \|U\|_2$, $M_V = \|V\|_2$, $M_W = \|W_o\|_2$, $M_x = \|X\|_F$, and $\rho = \|C'\|_2$. Then

$$\|\text{Quad}(X+\Delta X) - \text{Quad}(X)\|_F \le M_U M_V M_W\big(2\rho\, M_x\, \|\Delta X\|_F + M_x^2\, \|\Delta C'\|_2\big),$$

which proves the claim. $\qquad\square$

**Remarks.** (i) Degree normalization of $C'$ keeps $\rho = \|C'\|_2$ bounded across subjects/sites, tightening the stability constant. (ii) Results apply to signed $C$ and $C'$ and to piecewise-linear encoders (per linear region). (iii) If desired, a head-wise bound can add a $\sqrt{k}$ factor by $\|A \odot B\|_F \le \sqrt{k}\,\|A\|_F\|B\|_F$.

**Design choices justified by the propositions.**

- **Low-rank quadratic branch.** Prop. 1 shows that our Quadranet implements explicit degree-2 (edge–edge) interactions via separable bilinear terms. This directly motivates including the quadratic branch (to model interactions that MPNNs leave implicit) while keeping it *low rank* for efficiency.

- **Dual ASPP on features and connectivity.** Prop. 2 establishes that, under piecewise-linear encoders, each output depends on a degree-2 polynomial of entries of $C$ restricted to the *union of the two receptive fields*. This supports using $\mathcal{A}_{\text{feat}}$ and $\mathcal{A}_{\text{edge}}$ together to confine interactions to multi-scale neighborhoods—yielding motif-level sensitivity without deep message passing.

- **Choosing the rank $k$.** Prop. 3 characterizes the expressivity–efficiency tradeoff: the realized class equals cross-block quadratics of rank $\le k$ and approaches dense interactions as $k$ increases. This predicts a practical $k$-*sweep* (accuracy rises then saturates), guiding us to small $k$ that balances accuracy and cost.

- **Normalization for robustness.** Prop. 4 gives a Lipschitz bound in $X$ and $C'$ with constants depending on $\|C'\|_2$ and the low-rank factors. This justifies degree-normalizing $C'$ (to control $\|C'\|_2$) and preferring smaller $k$, both of which improve conditioning and cross-site robustness.

