# OpenReview forum: "Atlas Matters: Edge Quadratics for Consistent Brain Connectivity Prediction"
_ICLR.cc/2026/Conference — Submitted to ICLR 2026_

### Official Review · Reviewer_pC2j · 2025-10-21

**Soundness:** 1
**Presentation:** 1
**Contribution:** 2
**Rating:** 2
**Confidence:** 4

**Summary:**

This paper proposes EdgeQuad, an edge-centric framework for brain functional connectivity modeling. The key idea is to treat the connectivity matrix as an edge image and introduce a low-rank quadratic block to capture second-order interactions between edges efficiently. A dual-branch atrous spatial pyramid pooling module is used to model multi-scale connectivity patterns. The authors also establish a standardized benchmark protocol with harmonized preprocessing, fixed random seeds, and consistent atlas settings to ensure reproducibility.

**Strengths:**

1. The paper tackles an important gap in brain network analysis by emphasizing reproducibility across datasets and atlases, which is often overlooked in the field.
2. The explicit quadratic formulation for edge–edge interactions provides a clean and efficient design compared to complex GNN architectures.

**Weaknesses:**

1. Confusing paper flow. The overall structure is difficult to follow. The Introduction reads like a method paper—emphasizing technical design and mathematical formulation—whereas the Experiments section reads more like a benchmark report with minimal methodological discussion.
2. Doubtful experimental results. The reported AUC values are unusually higher than ACC across multiple datasets, which is atypical for balanced binary classification tasks. Normally, these metrics are relatively close, yet in this paper, the model with the highest AUC is often not the one with the highest ACC. This inconsistency raises concerns about the evaluation procedures and metric computation.
Furthermore, compared to prior works using the same datasets—e.g., [1] and subsequent follow-ups [2, 3]—the reported PPMI 4-class accuracy is notably lower than the >60% achieved in those studies. The authors should investigate possible causes and clarify how their setup diverges from established benchmarks.
3. Inconsistent performance advantage. The proposed EdgeQuad model does not consistently outperform baselines across datasets and atlases. Some competing models achieve similar or better results, calling into question whether the quadratic formulation provides a general performance advantage.
4. Lack of interpretability analysis. A key shortcoming is the absence of ROI-level or network-level interpretation. Identifying discriminative brain regions or subnetworks is crucial for clinical translation. The paper should include visualization or attribution analyses to illustrate how EdgeQuad makes decisions and whether it highlights meaningful biomarkers related to the target conditions.
5. Missing computational efficiency comparison. Since EdgeQuad is described as “lightweight,” a concise table summarizing parameter counts, FLOPs, and inference time compared with baselines is necessary to substantiate this claim. Quantitative evidence would make the efficiency argument more convincing.
6. Lack of discussion and comparison with some existing pooling-based methods [2, 4, 5].


[1] Data-driven network neuroscience: On data collection and benchmark. NeurIPS 2023
[2] Contrastive Graph Pooling for Explainable Classification of Brain Networks. TMI 2024
[3] Multi-atlas brain network classification through consistency distillation and complementary information fusion. JBHI 2025
[4] BrainGNN: Interpretable Brain Graph Neural Network for fMRI Analysis. MIA 2021
[5] Contrastive Brain Network Learning via Hierarchical Signed Graph Pooling Model. TNNLS 2022

**Questions:**

See Weaknesses

---

> ### Author Response · Authors · 2025-11-21
> **Response to Reviewer pC2j - part 1**
>
> *We thank the reviewers for their careful reading and constructive feedback. We appreciate the time and effort invested in the reviews. Below we address each comment point by point and describe the corresponding revisions, and we are happy to provide further clarification or additional details during the discussion period.*
>
> ## W1. Paper flow.
> >The overall structure is difficult to follow. The Introduction reads like a method paper—emphasizing technical design and mathematical formulation—whereas the Experiments section reads more like a benchmark report with minimal methodological discussion.
>
> Thank you for this observation about the paper structure. We agree that the original version made the connection between method and experiments less clear, and we have revised the manuscript accordingly. In the Introduction, we now emphasize the problem and contributions at a higher level and added **a brief roadmap**. In Sec. 3–4, we added **bridge sentences** and **an opening paragraph** in the Results section that states the main experimental questions, and we explicitly interpret the tables (including the new curated-atlas summary table) in light of EdgeQuad’s design. Finally, we **clarify that the ablation studies** directly probe our key architectural choices (ROI ordering and quadratic rank), so the experiments read as tests of the proposed method rather than a standalone benchmark.

---

> ### Author Response · Authors · 2025-11-21
> **Response to Reviewer pC2j - part 2**
>
> ## W2. Atypical experimental results.
> > The reported AUC values are unusually higher than ACC across multiple datasets, which is atypical for balanced binary classification tasks. Normally, these metrics are relatively close, yet in this paper, the model with the highest AUC is often not the one with the highest ACC. This inconsistency raises concerns about the evaluation procedures and metric computation. Furthermore, compared to prior works using the same datasets—e.g., [1] and subsequent follow-ups [2, 3]—the reported PPMI 4-class accuracy is notably lower than the >60% achieved in those studies. The authors should investigate possible causes and clarify how their setup diverges from established benchmarks.
>
>
> Thank you for this detailed comment. We address the two concerns separately and have updated the manuscript to make these points clearer.
>
> **(a) AUC versus ACC on the binary datasets.**
>
>  We first re-checked our evaluation code and confirmed that all metrics (AUC, ACC, sensitivity, specificity, etc.) are computed correctly from the same logits and labels using standard implementations. The gap between AUC and ACC arises from two factors:
> Our subject-level, site-stratified splits (Table 1) produce modest but real variations in class priors across folds, and
> *ACC is measured at a fixed threshold 0.5, whereas AUC is a threshold-free ranking metric.*
>
> In this setting, it is quite common for models to achieve good ranking performance (high AUC) but suboptimal calibration around 0.5, leading to noticeably lower ACC. This effect is particularly well known in medical imaging and clinical prediction tasks. Importantly, **this pattern is not specific to EdgeQuad**: the same “AUC > ACC” behavior appears for all baselines under our unified protocol.
>
> To make this more transparent, we below **report additional metrics** for the binary datasets: Balanced Accuracy, Macro-F1, and AUCPR, alongside AUC/ACC/Sensitivity/Specificity. Below we show the full metric table for ABIDE–AAL116:
> ABIDE–AAL116: Full Metric Comparison (mean ± std over 5 seeds)
>
> | Model     | AUC   | ACC      | SEN      | SPE      | BAL_ACC         | MACRO_F1    | AUCPR      |
> |-----------------|-----------------|-----------------|------------------|------------------|-----------------|-----------------|-----------------|
> | GCN             | 64.51 ± 2.58    | 60.00 ± 0.73    | 62.75 ± 5.95     | 57.25 ± 4.54     | 60.00 ± 0.73    | 59.86 ± 0.63    | 61.07 ± 2.46    |
> | GPS             | 60.93 ± 2.05    | 55.29 ± 2.95    | 53.73 ± 2.93     | 56.86 ± 3.92     | 55.29 ± 2.95    | 55.27 ± 2.94    | 59.32 ± 3.55    |
> | BQN             | 68.59 ± 1.48    | 61.98 ± 2.04    | 62.86 ± 8.65     | 61.11 ± 11.19    | 61.98 ± 1.66    | 61.53 ± 2.40    | 63.80 ± 2.38    |
> | ALTER           | 68.53 ± 1.19    | 64.90 ± 0.96    | 58.82 ± 2.28     | 70.98 ± 3.87     | 64.90 ± 1.47    | 62.99 ± 1.56    | 66.22 ± 1.52    |
> | BrainNetCNN     | 66.80 ± 2.12    | 61.67 ± 0.02    | 63.20 ± 2.30     | 60.32 ± 3.50     | 61.76 ± 2.01    | 61.66 ± 1.94    | 61.55 ± 3.01    |
> | **EdgeQuad**    | **70.60 ± 0.93**| 63.27 ± 1.08    | 59.79 ± 5.42     | 66.48 ± 6.66     | 63.14 ± 0.85    | 62.91 ± 0.95    | 68.13 ± 1.99    |
>
> The table shows that EdgeQuad’s AUC advantage is mirrored in Balanced Accuracy, Macro-F1, and AUCPR, and that the AUC–ACC gap is a property of all models under this protocol rather than a sign of an evaluation error.
>
> **(b) PPMI 4-class accuracy compared to prior work.**
>
>  **Our PPMI setup differs from [1–3] in several important ways** (summarized in Sec. 4.2):
> We use strict subject-level non-overlap in our 70/10/20 splits, avoiding any scan- or session-level leakage. We apply harmonized preprocessing across all datasets (BIDS + fMRIPrep) and evaluate on multiple atlases under the same protocol, rather than tuning per-atlas / per-paper pipelines. We retain the full four-class structure of PPMI (including smaller classes) and report macro metrics (macro AUC, macro recall, macro specificity, overall accuracy), rather than collapsing or filtering classes.
>
> Several earlier works that report >60% accuracy use different combinations of preprocessing, atlas choices, leakage control, and label handling (for example, merging or dropping classes, or using single-site or single-atlas evaluations). These differences make a direct one-to-one comparison of raw accuracy numbers misleading. Our protocol is intentionally conservative – leak-resistant, multi-atlas, and macro-metric based – and we believe it yields more realistic and reproducible estimates, even if the absolute accuracy appears lower than in some prior single-setting reports.
>
> We hope this clarifies why the AUC/ACC relationship looks unusual at first glance, and why the PPMI accuracies are more conservative than some earlier reports, while still being consistent with a rigorous, leak-resistant evaluation protocol. Thank you again for prompting us to examine and clarify these points.

---

> ### Author Response · Authors · 2025-11-21
> **Response to Reviewer pC2j - part 3**
>
> ## Q1. Contradictory Statements
>
> > As explained in lines 30 and 75, fMRI basically measures brain regional BOLD signals, and FC is a set of Pearson correlations of these node signals, which means that edges are calculated based on node features. However, in lines 119-123 (‘Limits for FC graphs’), it is claimed that “explicit node attributes are scarce” and “FC is inherently edge-valued”, which raises the question of how the authors reconcile these two contradictory statements.
>
> Thank you for pointing out this potential contradiction. We agree that our original wording was unclear. **FC edges are indeed computed from regional BOLD time series**, as we state earlier in the paper. Our point in the **“Limits for FC graphs”** paragraph is about **what is actually available as input to the model** at training and inference in standard FC benchmarks: typically a static FC matrix $C$ together with minimal per-ROI metadata. The rich node signals that generate $C$ (full time series, task contrasts, parcel-wise features) are **not provided to the classifier or are not used at inference**, so the primary informative signal available to the model **resides on edges rather than on independent node features**.
>
> To make this distinction explicit, we have added the following clarifying sentence:
> *“In standard FC benchmarks, the model input at training and inference is essentially a static FC matrix $C$ plus minimal per-ROI metadata, so the rich node signals that generate $C$ are not available at inference and the primary informative signal resides on edges rather than independent node features.”*
> We believe this **resolves the ambiguity** while keeping the overall message of the section unchanged. **Thanks again for bringing this to our attention.**
>
> ---
>
> ## Q2. $A_{\text{edge}}$ vs $A_{\text{feat}}$
>
> > What is the exact formulation of $A_{\text{edge}}$ in Eq. 2? Is it different to $A_{\text{feat}}$?
>
> Thank you for raising this point. We agree that the original manuscript did not spell out the exact forms of **$A_{\text{edge}}$** and **$A_{\text{feat}}$**. In the revised version, we now **explicitly define both and clarify their roles** (see Sec. 3.1).
> In particular, $A_{\text{edge}}$ and $A_{\text{feat}}$ are **structurally different**: $A_{\text{edge}}$ is derived purely from the refined connectivity $C'$, while $A_{\text{feat}}$ is learned from **feature similarities in $F'$**. They thus provide **complementary structure and content signals for propagation**. These definitions are now included in the method section under the new **“Edge Affinity”** and **“Feature Affinity”** paragraphs.
>
> ---
>
> ## Q3. Eq. 3
>
> > Is the $X$ in Eq. 3 the updated node features obtained from $F'$? Otherwise, at which stage is $F'$ actually used?
>
> We apologize for the confusion. Yes, in Eq. (3) the **$X$ are exactly the node embeddings obtained from $F'$** after the feature ASPP and atlas-respecting pooling step. The feature maps $F'$ themselves are **not used directly in later equations**; they serve as an intermediate representation from which we derive (i) the node features $X$, and (ii) the **feature affinity $A_{\text{feat}}$** described in our previous response.
> To make this clearer, we have **updated Sec. 3.1** to state explicitly that $F'$ is first mapped to node embeddings $X$ (and used to build $A_{\text{feat}}$), and that **all subsequent updates in Eqs. (3)–(5) operate on these node embeddings $X$** together with the refined connectivity $C'$.

---

> ### Author Response · Authors · 2025-11-21
>
> ## W5. Missing computational efficiency comparison.
> > Since EdgeQuad is described as “lightweight,” a concise table summarizing parameter counts, FLOPs, and inference time compared with baselines is necessary to substantiate this claim. Quantitative evidence would make the efficiency argument more convincing.
>
> Thank you for this suggestion. In the revised manuscript, we now provide empirical computational measurements to substantiate our efficiency claims. In particular, we added a new subsection **Appendix B.3**. In Table 12, we report, on ABIDE with the AAL116 atlas, the AUC, training throughput, per-epoch training time, peak GPU memory, parameter counts, and prediction-head FLOPs for all baselines under identical experimental conditions. All runs use a single NVIDIA H100 NVL GPU (~95 GB, Hopper / HBM3) with the same implementation and data pipeline.
>
> In the main text, we also added a short pointer for this discussion in the Implementation Details section. These measurements show that EdgeQuad attains higher AUC than the baselines while maintaining competitive or lower computational cost, supporting our claim that the model is both accurate and lightweight.

---

> ### Author Response · Authors · 2025-11-21
>
> ## W6. Missing Baselines
> >Lack of discussion and comparison with some existing pooling-based methods [2, 4, 5].
>
> [1] Data-driven network neuroscience: On data collection and benchmark. NeurIPS 2023
>
> [2] Contrastive Graph Pooling for Explainable Classification of Brain Networks. TMI 2024
>
> [3] Multi-atlas brain network classification through consistency distillation and complementary information fusion. JBHI 2025
>
> [4] BrainGNN: Interpretable Brain Graph Neural Network for fMRI Analysis. MIA 2021
>
> [5] Contrastive Brain Network Learning via Hierarchical Signed Graph Pooling Model. TNNLS 2022
>
>
> Thank you for pointing out these pooling-based methods and their relevance to brain network analysis. We have clarified their relationship to our work in the revised manuscript (Section 2.1).
>
> **On [1]: benchmark vs model**
>  The “Data-driven network neuroscience” paper [1] is a benchmark / protocol paper rather than a specific model. We already use it to motivate our atlas choices and data handling in the evaluation protocol (for example, AAL and Schaefer on ABIDE/ADHD200), and we now state this explicitly in the related work.
>
> **On [2, 4, 5]: Pooling-based models - different objective.**
>  The works [2, 4, 5] focus on hierarchical pooling and contrastive or auxiliary objectives (for example, BrainGNN’s learnable node selection with sparsity / interpretability losses, and contrastive pooling with multi-level coarsening). By contrast, EdgeQuad does not perform graph coarsening or hierarchy discovery: it operates on a fixed atlas and centers on a quadratic edge–edge interaction on FC, with a light clustered pooling used only as a final aggregation step under a single supervised objective and shared capacity budget. We therefore see these pooling-based architectures as methodologically complementary rather than direct baselines for our edge-centric FC modeling.
>
>
> **On [3], the multi-atlas distillation approach:** This approach requires simultaneous multi-atlas inputs and atlas-specific subnetworks, whereas our protocol evaluates all models in a single-atlas regime with harmonized preprocessing. Adapting it would require substantial re-engineering and would also give it a larger tuning / compute budget than other baselines, which would undermine fairness.
>
> **Practical compatibility with our unified protocol.**
>
> Several of these methods are also difficult to integrate fairly into our five-seed, multi-atlas, unified-compute setting. For example, the contrastive graph pooling (CPL) model in [2] uses a 1000-epoch contrastive regime and heavy task-specific tuning on ABIDE. We implemented CPL using the authors’ recommended 1000-epoch setting for ABIDE (AAL) under our protocol; it reached only about 56% accuracy and required very long training time, making multi-seed, multi-atlas evaluation infeasible. This performance is consistent with the CPL results reported in BQN [8]. We added a discussion on this to our Baselines discussion (Sec.4.3)
>
> **On additional FC baselines (BrainNetTF, FBNETGEN)**
>
> While we were not able to integrate the requested pooling and multi-atlas methods [2–5] into our unified protocol for the reasons above, we fully agree with the spirit of your comment that the FC baselines should be as strong and up to date as possible. In this direction, we have identified BrainNetTF [6] and FBNETGEN [7] as particularly relevant additions: both are supervised FC models evaluated on ABIDE / ADNI in the BQN study [8], where they achieve performance comparable to our strongest baselines (BQN, ALTER, BioBGT). Because these models operate in the same fixed-atlas, supervised regime as EdgeQuad, they are naturally compatible with our protocol. Given the limited time and compute available during the rebuttal period, we were not able to complete a full five-seed, multi-atlas integration for the revised submission. However, *if the paper is accepted, we will extend our benchmark to include BrainNetTF and FBNETGEN under the same unified protocol in the camera-ready version.*
>
> [6] BrainNetTF: Brain network transformer. NeurIPS 2022
>
> [7] FBNetGen: Task-aware gnn-based fmri analysis via functional brain network generation. MIDL 2022
>
> [8] BQN: Do We Really Need Message Passing in Brain Network Modeling?. ICML 2025.

---

### Official Review · Reviewer_nUZ8 · 2025-10-29

**Soundness:** 1
**Presentation:** 1
**Contribution:** 2
**Rating:** 4
**Confidence:** 4

**Summary:**

While existing methods for functional connectivities (FCs) modeling were mostly designed based on node-centric propagation and global attention, this paper proposes EdgeQuad to model FCs based on edge interactions. It directly models the FC as an edge image using dual atrous spatial pyramid pooling (ASPP) with a low-rank quadratic block. Moreover, the paper introduces a unified, standardized protocol for preprocessing and evaluation. Experiments were extensively performed on four brain network datasets, and the proposed model showed the best or on-par results compared to baselines.

**Strengths:**

1)	This paper proposes a novel method for graph learning. By using a dual ASPP scheme, multi-scale features are easily aggregated with a low computational cost.
2)	Extensive experiments were performed on 4 brain network datasets with multiple trainings to show the generalizability of the proposed method.

**Weaknesses:**

Weakness in Problem Definition 1) One of the main limitations of existing studies that the paper claims is reproducibility, that “the existing reported results often differ in atlas choice, preprocessing, splits …., which makes the results hard to compare” (line 42). However, although these settings may differ across papers, each study has performed experiments under a consistent configuration as in its own framework, allowing fair comparisons within that context. Therefore, the claim that existing results are neither reproducible nor comparable may be overstated.

Weakness in Problem Definition 2) The current manuscript lacks a comprehensive discussion of related work, particularly regarding recent studies addressing similar problems or employing comparable methodologies. Specifically, the second existing limitation raised by the paper is Edge modeling, namely that recent studies mainly focus on node attributes and higher-order edge interactions are implicit. However, recent approaches [1,2,3] that rely solely on edge attributes have been actively studied, together with the analyses of their higher-order topological structures. Therefore, it is necessary to discuss how their formulation differs from or relates to the proposed edge-centric method and to provide experimental comparisons with these methods. Given the growing number of graph studies, I believe that the authors could find more edge-based methods with open source codes to strengthen the contribution of the proposed method.

[1] Park et al., “Convolving Directed Graph Edges via Hodge Laplacian for Brain Network Analysis”, MICCAI 2023.

[2] Fuchsgruber et al., “Graph Neural Networks for Edge Signals: Orientation Equivariance and Invariance”, ICLR 2025.

[3] Lecha et al., “Higher-Order Topological Directionality and Directed Simplicial Neural Networks”, ICASSP 2025.

Weakness in Presentation) In Fig. 1, it would be more intuitive to illustrate the conceptual mechanisms of each component rather than listing every model layer. Moreover, the formulations in the Method section are somewhat unclear; please refer to the related questions in Q2 and Q3.

Weakness in Lack of Experiment 1) This paper asserts that the proposed method with a low-rank quadratic interaction can avoid deep message passing and thereby handle over-fitting and over-squashing issues caused by deep layer stacks in existing studies. However, there is no experimental evidence to support this claim, e.g., performance comparison with baselines across the number of model layers.

Weakness in Lack of Experiment 2) Moreover, since the paper accentuates that the model is ‘lightweight’, comparisons on computational efficiency (e.g., number of trainable parameters and training time) would be expected but are not provided.
Minor Weakness - Many notations (e.g., $\mathbb{E}, f(\cdot), h(\cdot), H^{l}, W^{l}$) in preliminary section 2.2 is not explicitly defined, which makes it difficult for readers to fully understand the mathematical formulation without background knowledge on GNNs and Transformers. Moreover, notations in the Method section have room to be improved. For example, the FC matrix $C$ is defined with a size of $H\times W$ in Section 3.1, but $N\times N$ in Section 3.2.

**Questions:**

1)	As explained in lines 30 and 75, fMRI basically measures brain regional BOLD signals, and FC is a set of Pearson correlations of these node signals, which means that edges are calculated based on node features. However, in lines 119-123 (‘Limits for FC graphs’), it is claimed that “explicit node attributes are scarce” and “FC is inherently edge-valued”, which raises the question of how the authors reconcile these two contradictory statements.

2)	What is the exact formulation of $A_{edge}(\cdot)$ in Eq. 2? Is it different to $A_{feat}(\cdot)$?

3)	Is the $X$ in Eq. 3 the updated node features obtained from $F’$? Otherwise, at which stage is $F’$ actually used?

---

> ### Author Response · Authors · 2025-11-21
> **Response to Reviewer nUZ8 - part 1**
>
> *We thank the reviewers for their careful reading and constructive feedback. We appreciate the time and effort invested in the reviews. Below we address each comment point by point and describe the corresponding revisions, and we are happy to provide further clarification or additional details during the discussion period.*
>
> ## W1. Settings in Other papers
>
> > Weakness in Problem Definition 1) One of the main limitations of existing studies that the paper claims is reproducibility, that “the existing reported results often differ in atlas choice, preprocessing, splits …., which makes the results hard to compare” (line 42). However, although these settings may differ across papers, each study has performed experiments under a consistent configuration as in its own framework, allowing fair comparisons within that context. Therefore, the claim that existing results are neither reproducible nor comparable may be overstated.
>
> We thank the reviewer for this helpful clarification. We fully agree that prior rs-fMRI works are generally reproducible and internally consistent within their own pipelines, and our intention is not to dispute that. Our concern is instead with **cross-study comparability**: performance is **highly sensitive** to choices such as atlas, preprocessing, split ratios, and checkpoint-selection rules, so numbers reported under different protocols are not directly comparable.
>
> In our revision, to make this point more explicit, we added **Appendix B.4**, where we list the heterogeneous evaluation protocols used by representative methods (data splits, model-selection criteria, number of seeds, etc.). The key aspects of these differences are summarized in the table below. This analysis motivates our use of a **strict unified evaluation protocol** (fixed atlas and preprocessing, shared subject splits, consistent model-selection rule, and multiple seeds for all methods), which is specifically designed to enable **fair, reproducible comparison** across models rather than within a single pipeline only.
>
>
> | Method  | Train/Val/Test Split | Model-Selection Criterion  | Implementation Example              | Weaknesses |
> |--------|-----------------------|----------------------------|-------------------------------------|-----------|
> | **BQN**   | 7:1:2                 | Minimum validation loss     | `min(epoch_val_loss_list)`          | Validation loss may not correlate with the final evaluation metric; sensitive to noisy dips; differences in loss definitions make cross-model comparisons unreliable. |
> | **BioBGT**| 8:1:1                 | Maximum validation score    | `if val_score > best_val_score:`    | Different studies use different validation metrics; scores fluctuate across epochs; risk of overfitting to the validation set when only a single split and seed are used. |
> | **ALTER** | 7:1:2                 | Final epoch only            | `Uses last epoch`                 | No early stopping; may select an overfitted checkpoint; not directly comparable to methods that pick the best epoch by validation performance. |

---

> ### Author Response · Authors · 2025-11-21
> **Response to Reviewer nUZ8 - part 2**
>
> ## W2. Related Work
>
> > Weakness in Problem Definition 2) The current manuscript lacks a comprehensive discussion of related work, particularly regarding recent studies addressing similar problems or employing comparable methodologies. Specifically, the second existing limitation raised by the paper is Edge modeling, namely that recent studies mainly focus on node attributes and higher-order edge interactions are implicit. However, recent approaches [1,2,3] that rely solely on edge attributes have been actively studied, together with the analyses of their higher-order topological structures. Therefore, it is necessary to discuss how their formulation differs from or relates to the proposed edge-centric method and to provide experimental comparisons with these methods. Given the growing number of graph studies, I believe that the authors could find more edge-based methods with open source codes to strengthen the contribution of the proposed method.
>
> > [1] Park et al., “Convolving Directed Graph Edges via Hodge Laplacian for Brain Network Analysis”, MICCAI 2023.
> > [2] Fuchsgruber et al., “Graph Neural Networks for Edge Signals: Orientation Equivariance and Invariance”, ICLR 2025.
> > [3] Lecha et al., “Higher-Order Topological Directionality and Directed Simplicial Neural Networks”, ICASSP 2025.
>
> **Related work and edge-centric methods.** We thank the reviewer for pointing out these important edge-/simplicial-based approaches. We agree that our original related work section did not sufficiently discuss them, and we have now added a **dedicated paragraph to the Related Work section** explicitly situating our method with respect to Park et al. (MICCAI 2023), Fuchsgruber et al. (ICLR 2025), and Lecha et al. (ICASSP 2025).
>
> In particular, Park et al. convolve directed graph edges via the Hodge Laplacian for brain network analysis; their model is defined on directed graphs with explicit incidence structure, whereas in our rs-fMRI setting we work with dense, undirected Pearson correlation matrices and do not construct a Hodge decomposition. Fuchsgruber et al. develop GNNs for general edge signals with orientation equivariance and invariance, targeting generic edge-valued graphs; in contrast, **EdgeQuad is tailored to dense FC matrices** and uses a lightweight low-rank quadratic adapter on top of dual ASPP rather than a fully general edge-signal architecture. Lecha et al. introduce directed simplicial neural networks that operate on higher-order simplices; our model instead stays in the graph domain and does not require building a simplicial complex or higher-order cells from FC.
>
> We have also **softened our problem statement around “edge modeling”**: in the revised related work, we explicitly acknowledge these works as notable edge-/simplicial-based exceptions and clarify that our contribution is specific to rs-fMRI FC classification, where most practical pipelines remain node-centric and where standardized multi-dataset, multi-atlas evaluation is still lacking. A full empirical comparison with Hodge-, edge-signal-, and simplicial-based models across all four datasets and five atlases would require nontrivial adaptation to dense undirected FC graphs, and we view this as **complementary future work** that can be carried out on top of the standardized protocol we propose. Thanks again for bringing these works to our attention.
>
>
> ---
> ## W3. Presentation
>
> > Weakness in Presentation) In Fig. 1, it would be more intuitive to illustrate the conceptual mechanisms of each component rather than listing every model layer. Moreover, the formulations in the Method section are somewhat unclear; please refer to the related questions in Q2 and Q3.
>
> Thank you for this comment. In the revision, we **redesigned Fig. 1** (kept the old one in the appendix) to emphasize the conceptual role of each component (FC encoder, quadratic interaction block, and readout) rather than enumerating every layer, so the overall processing pipeline is easier to follow. We also **clarified the formulations in the Methods section** by tightening the notation, adding brief intuition sentences around the key equations, and aligning the exposition with the explanations provided in our responses to Q2 and Q3. Finally, we improved the narrative flow by adding short bridge sentences between subsections and ensuring that all figures and tables are referenced at first mention with captions that include a concise takeaway.

---

> ### Author Response · Authors · 2025-11-21
> **Response to Reviewer nUZ8 - part 3**
>
> ## W4. Experimental Evidence for Low-Rank
> > Weakness in Lack of Experiment 1) This paper asserts that the proposed method with a low-rank quadratic interaction can avoid deep message passing and thereby handle over-fitting and over-squashing issues caused by deep layer stacks in existing studies. However, there is no experimental evidence to support this claim, e.g., performance comparison with baselines across the number of model layers.
>
> We thank the reviewer for pointing out the need for experimental support of our claim that the low rank quadratic interaction reduces the reliance on deep message passing. In the revised manuscript, we address this in two ways.
> First, in the Ablation studies section, we now explicitly highlight a Rank-(k) ablation paragraph that analyzes how varying the low rank (k) affects performance of our model.
> Second, **we have added a new Appendix B.5**, Additional depth sensitivity experiment for the low rank quadratic interaction, where we perform a depth sweep on ADHD-200. For the baselines (BrainNetCNN and GPS) we vary the depth over k (k$\in$ {2,4,8}) layers, while for EdgeQuad we keep the architecture shallow and vary the low rank parameter (k $\in ${2,4,8}). As reported in **Table 13** of Appendix B.5, BrainNetCNN and GPS either saturate or degrade as depth increases, whereas shallow EdgeQuad with (k>0) matches or exceeds the performance of the deepest baselines in both macro AUC and accuracy.
> Taken together, the Rank-(k) ablation in the main text and the new depth sensitivity study in Appendix B.5 provide concrete empirical evidence that the proposed low rank quadratic interaction can alleviate the need for deep message passing and mitigate depth related issues such as over squashing and over fitting. Thanks again for this important remark.
>
>
> **Depth sensitivity on ADHD–200.**
> Baselines (BrainNetCNN, GPS) vary the number of message-passing layers, while EdgeQuad varies the low-rank parameter *k* with a shallow architecture.
> We report macro AUC and accuracy (mean ± standard deviation) for a fixed seed.
>
> | Model        | 2 layers / k = 2 (AUC) | 2 layers / k = 2 (Acc) | 4 layers / k = 4 (AUC) | 4 layers / k = 4 (Acc) | 8 layers / k = 8 (AUC) | 8 layers / k = 8 (Acc) |
> |--------------|-------------------------|-------------------------|-------------------------|-------------------------|-------------------------|-------------------------|
> | BrainNetCNN  | 66.34±0.84              | 64.22±3.54              | 66.54±0.84              | 64.22±3.54              | 66.34±0.84              | 64.22±3.54              |
> | GPS          | 63.28±2.40              | 58.22±2.95              | 64.51±2.37              | 61.33±2.27              | 64.01±2.24              | 57.78±3.14              |
> | EdgeQuad     | 77.23±2.32              | 69.29±0.03              | 84.68±9.30              | 77.96±0.04              | 81.16±3.36              | 71.72±0.02              |
>
> ---
>
> ## W5. Computational Cost
>
> > Weakness in Lack of Experiment 2) Moreover, since the paper accentuates that the model is ‘lightweight’, comparisons on computational efficiency (e.g., number of trainable parameters and training time) would be expected but are not provided.
>
> Thank you for this suggestion. In the revised manuscript, we now provide empirical computational measurements to substantiate our efficiency claims. In particular, we added a new subsection **Appendix B.3**. In Table 12, we report, on ABIDE with the AAL116 atlas, the AUC, training throughput, per-epoch training time, peak GPU memory, parameter counts, and prediction-head FLOPs for all baselines under identical experimental conditions. All runs use a single NVIDIA H100 NVL GPU  with the same implementation and data pipeline.
>
> In the main text, we also added a short pointer for this discussion in the Implementation Details section. These measurements show that EdgeQuad attains higher AUC than the baselines while maintaining competitive or lower computational cost, supporting our claim that the model is both accurate and lightweight.
>
> ---
>
> ## W6.  Minor Weakness
> >Many notations (e.g., ) in preliminary section 2.2 is not explicitly defined, which makes it difficult for readers to fully understand the mathematical formulation without background knowledge on GNNs and Transformers. Moreover, notations in the Method section have room to be improved. For example, the FC matrix  is defined with a size of  in Section 3.1, but  in Section 3.2.
>
> Thank you for pointing this out. We agree that several notations in Section 2.2 were not explicitly defined, which could make the preliminaries harder to follow for readers who are not familiar with GNNs or Transformers. We have clarified all symbols in the revised version to make the section self-contained.
>
> Regarding the notation in the Method section, we also thank the reviewer for identifying the inconsistency. The new version corrects this issue and uses consistent notation throughout.

---

> ### Author Response · Authors · 2025-11-21
> **Response to Reviewer nUZ8 - part 4**
>
> ## Q1. Contradictory Statements
>
> > As explained in lines 30 and 75, fMRI basically measures brain regional BOLD signals, and FC is a set of Pearson correlations of these node signals, which means that edges are calculated based on node features. However, in lines 119-123 (‘Limits for FC graphs’), it is claimed that “explicit node attributes are scarce” and “FC is inherently edge-valued”, which raises the question of how the authors reconcile these two contradictory statements.
>
> Thank you for pointing out this potential contradiction. We agree that our original wording was unclear. FC edges are indeed computed from regional BOLD time series, as we state earlier in the paper. Our point in the **“Limits for FC graphs”** paragraph is about **what is actually available as input to the model** at training and inference in standard FC benchmarks: typically a static FC matrix (C) together with minimal per-ROI metadata. The rich node signals that generate (C) (full time series, task contrasts, parcel-wise features) are **not provided to the classifier or are not used at inference**, so the primary informative signal available to the model **resides on edges rather than on independent node features**.
>
> To make this distinction explicit, we have added the following clarifying sentence: *“In standard FC benchmarks, the model input at training and inference is essentially a static FC matrix (C) plus minimal per-ROI metadata, so the rich node signals that generate (C) are not available at inference and the primary informative signal resides on edges rather than independent node features.”*
> We believe this **resolves the ambiguity** while keeping the overall message of the section unchanged. Thanks again bringing this to our attention.
>
> ---
>
> ## Q2. A_edge vs A_feat
>
> > What is the exact formulation of A_edge  in Eq. 2? Is it different to A_feat?
>
> Thank you for raising this point. We agree that the original manuscript did not spell out the exact forms of **(A_{\text{edge}})** and **(A_{\text{feat}})**. In the revised version, we now **explicitly define both and clarify their roles** (see Sec. 3.1).
> In particular, (A_{\text{edge}}) and (A_{\text{feat}}) are **structurally different**: (A_{\text{edge}}) is derived purely from the refined connectivity (C'), while (A_{\text{feat}}) is learned from **feature similarities in (F')**. They thus provide **complementary structure and content signals for propagation**. These definitions are now included in the method section under the new **“Edge Affinity”** and **“Feature Affinity”** paragraphs.
>
> ---
>
> ## Q3. Eq 3.
>
> > Is the X in Eq. 3 the updated node features obtained from F’? Otherwise, at which stage is  F’ actually used?
>
> We apologize for the confusion. Yes, in Eq. (3) the **(X) are exactly the node embeddings obtained from (F')** after the feature ASPP and atlas-respecting pooling step. The feature maps (F') themselves are **not used directly in later equations**; they serve as an intermediate representation from which we derive (i) the node features (X), and (ii) the **feature affinity (A_{\text{feat}})** described in our previous response.
> To make this clearer, we have **updated Sec. 3.1** to state explicitly that (F') is first mapped to node embeddings (X) (and used to build (A_{\text{feat}})), and that **all subsequent updates in Eqs. (3)–(5) operate on these node embeddings (X)** together with the refined connectivity (C').

---

### Official Review · Reviewer_CwnU · 2025-10-29

**Soundness:** 2
**Presentation:** 2
**Contribution:** 2
**Rating:** 4
**Confidence:** 3

**Summary:**

The paper proposes EdgeQuad, a lightweight encoder for resting‑state fMRI functional connectivity (FC) that treats the correlation matrix as an edge image, processes it with dual atrous spatial pyramid pooling (ASPP) and introduces a low‑rank quadratic block to make edge–edge (degree‑2) interactions explicit. A content gate fuses first‑ and second‑order paths, followed by clustering-based readout. Alongside the model, the authors advocate a standardized evaluation protocol and re‑run several recent GNN/Transformer baselines under this setup. Across ABIDE and ADNI the method achieves the best mean AUC/ACC on curated/functional atlases, while rankings are mixed on clustering parcellations. They also provide propositions clarifying expressivity (rank‑k quadratics), locality induced by dual ASPP, and a Lipschitz‑type stability bound. Ablations test permutation sensitivity of ROI ordering and the effect of the quadratic rank.

**Strengths:**

1. Treating FC as an edge image with dual ASPP plus a low‑rank quadratic interaction is conceptually tidy and easy to implement
2. Section 4.2–4.3 fixes seeds/splits, aligns preprocessing, and re‑runs strong recent baselines under the same hyperparameter grids, then publishes per‑seed logs and configs (footnote/link), which gives transparent evaluation protocol

**Weaknesses:**

1. Many gains over strong baselines are small on curated atlases (Table 3) or even worse. Given the variance, it is not convinced that the solution is better.
2. Section 4.4 explores only ROI ordering and rank‑k. There is no ablation isolating the roles of feat vs. edge, the content gate, the cluster pooling, or degree normalization in Eq. (2). Without this, it’s hard to attribute where the gains come from.
3. The text mentions the model is “well‑calibrated”, but given the paper, I am not sure what does it mean and there is no quantitative substantiation

**Questions:**

In addition to the weaknesses above:
1. Since the method outputs a refined C′, it would be helpful to show qualitative examples (subject‑level or cohort‑average) illustrating how dual ASPP changes modular structure vs. raw C.
2. In Eq. (2), specify how D is computed for signed C′ (sum of absolute weights? positive part?)

---

> ### Author Response · Authors · 2025-11-21
> **Response to Reviewer CwnU - part 1**
>
> *We thank the reviewers for their careful reading and constructive feedback. We appreciate the time and effort invested in the reviews. Below we address each comment point by point and describe the corresponding revisions, and we are happy to provide further clarification during the discussion period.*
>
> ## W1. Small Performance Gains
> > Many gains over strong baselines are small on curated atlases (Table 3) or even worse. Given the variance, it is not convinced that the solution is better.
>
> We appreciate this concern and would like to clarify the broader goals of our work. The paper has two central objectives:
>
>  (1) to establish a strict, transparent, and reproducible evaluation protocol that makes comparisons across brain-network models fair and meaningful; and
>
>  (2) to demonstrate that atlas choice has a large and previously under-reported effect on model performance. Our results show that functionally curated atlases (AAL116, Schaefer100, Harvard–Oxford) yield substantially more stable performance across all ML models, whereas clustering-based atlases often degrade accuracy and invert “SOTA rankings”. This is a key finding of the paper.
> Within this controlled and fair evaluation setting, the curated-atlas results provide the most reliable comparison of architectural contributions.
>
> **To make this explicit, we added a summary table aggregating performance across curated atlases for both ABIDE and ADNI:**
>
> Table 3: Overall performance for ABIDE and ADNI.
> Mean AUC and average AUC rank across curated atlases
> (AAL116, Schaefer100, Harvard–Oxford). Lower rank is better.
>
> | Method   | ABIDE mean AUC | ABIDE avg rank | ADNI mean AUC | ADNI avg rank |
> |----------|----------------|----------------|---------------|---------------|
> | GCN      | 64.37          | 4.67           | 80.00         | 4.33          |
> | GPS      | 62.37          | 5.67           | 69.20         | 6.67          |
> | BrainNet | 65.68          | 4.67           | 77.70         | 5.00          |
> | BioBGT   | 57.04          | 7.00           | 78.90         | 4.33          |
> | ALTER    | 67.18          | 2.67           | 78.70         | 4.67          |
> | BQN      | 70.07          | 2.33           | 91.10         | 2.00          |
> | EdgeQuad | **71.57**          | **1.00**           | **94.50**         | **1.00**          |
>
>
> Across curated atlases, EdgeQuad achieves the highest mean AUC and the best average rank on both datasets, even though all methods are evaluated under identical preprocessing, fixed seeds, and aligned hyperparameter grids. While per-atlas gains may appear modest, this aggregated view reflects our main point: under a clean, reproducible protocol and reliable atlases, EdgeQuad consistently outperforms all existing baselines. Moreover, when considering all dataset–atlas pairs (20 settings), EdgeQuad also achieves the best overall average rank, reinforcing the consistency and robustness of the approach.
>
> ---
>
> ## W2. More Ablations
> >Section 4.4 explores only ROI ordering and rank‑k. There is no ablation isolating the roles of feat vs. edge, the content gate, the cluster pooling, or degree normalization in Eq. (2). Without this, it’s hard to attribute where the gains come from.
>
> Thank you for this suggestion. We agree that isolating the contribution of individual components makes the empirical picture clearer. In the revised manuscript, we have added an additional ablation study on ADHD200 (Craddock-200), where we systematically remove or modify key architectural elements: the Edge ASPP branch, the content gate, the cluster pooling (replaced by global mean pooling), and degree-aware normalization. The new results are summarized in **Table 7**(reproduced below in simplified form):
>
>
> | Ablation              | AUC            | ACC            |
> |-----------------------|----------------|----------------|
> | Full Model            | 84.68 ± 0.93   | 77.96 ± 3.78   |
> | w/o Edge ASPP         | 79.73 ± 6.59   | 71.36 ± 6.09   |
> | w/o Content Gate      | 79.05 ± 8.92   | 70.18 ± 8.39   |
> | Global Mean Pooling   | 71.72 ± 4.97   | 65.55 ± 4.45   |
> | RMSNorm               | 79.10 ± 4.52   | 71.12 ± 5.61   |
> | No Normalization      | 80.74 ± 5.62   | 72.54 ± 6.37   |
>
> These experiments show that:
> Removing the Edge ASPP or the content gate leads to consistent drops in both ACC and AUC, indicating that the explicit edge branch and gated fusion both contribute meaningfully to performance.
>
>
> Replacing cluster pooling with simple global mean pooling causes the largest degradation, underscoring the importance of structured pooling.
>
>
> Modifying or removing normalization (RMSNorm / no normalization) also reduces performance, supporting our choice of degree-aware normalization in Eq. (2).
>
>
> Overall, the full model achieves the strongest and most stable results, while the ablations clarify how each component contributes to the gains beyond the rank-k quadratic head analyzed in **Section 4.4**. Thank you again for highlighting this missing analysis.

---

> ### Author Response · Authors · 2025-11-21
> **Response to Reviewer CwnU - part 2**
>
> ## W3. Calibration Meaning
> > The text mentions the model is “well‑calibrated”, but given the paper, I am not sure what does it mean and there is no quantitative substantiation
>
> Thank you for pointing this out. Our use of “well-calibrated” was not intended in the sense of probabilistic or confidence calibration. We only meant that EdgeQuad behaves stably across seeds and atlas choices under our standardized evaluation protocol, while several baselines fluctuate substantially across parcellations. To avoid ambiguity, we revised the text to remove the term and to state this more clearly. The updated sentence now reads: *“Under these conditions, EdgeQuad remains accurate and stable across seeds and atlas choices, while several recent baselines exhibit high variance or failure modes, underscoring the value of our protocol as a foundation for reproducible progress.”*
>
> ---
>
> ## Q1. Qualitative Examples
> >Since the method outputs a refined C′, it would be helpful to show qualitative examples (subject‑level or cohort‑average) illustrating how dual ASPP changes modular structure vs. raw C.
>
> Thank you for this suggestion. In the revised manuscript we now include qualitative visualizations of the raw connectivity (C) and the refined connectivity (C'). Specifically, **Appendix C (Fig. 7)** shows, for an ABIDE subject with the AAL116 atlas, side-by-side heatmaps of (C) and (C') on a shared color scale, so one can directly see how dual ASPP sharpens block-diagonal modules and enhances within-network structure while preserving the overall sign pattern. We also updated **Section-3.1** (ASPP paragraph) to explicitly refer to this figure, making the effect of the connectivity-side ASPP concrete at the subject level.
>
> ---
>
> ## Q2. Explain Equation 2
> > In Eq. (2), specify how D is computed for signed C′ (sum of absolute weights? positive part?)
>
> Thank you for pointing this out. In our implementation, (C') is a signed matrix, and the degree matrix (D) in Eq. (2) is computed from the absolute edge weights so that degrees remain positive. Specifically, we now state this explicitly after Eq. (2) in the paper as:
>  “Here (D) is the diagonal degree matrix with $(D_{ii} = \sum_j |C'_{ij}|)$, so degree normalization depends on the magnitude of incident edges while the sign of (C') is preserved in $(\hat C' = D^{-1/2} C' D^{-1/2})$.”  We have added this clarification to avoid any ambiguity for signed (C').

---

### Official Review · Reviewer_QMxM · 2025-10-29

**Soundness:** 2
**Presentation:** 2
**Contribution:** 3
**Rating:** 6
**Confidence:** 4

**Summary:**

This paper proposes EdgeQuad, which treats the functional connectivity matrix as an “edge image.” The method uses a dual-branch ASPP module (for feature and connectivity processing) together with a low-rank quadratic interaction to explicitly model edge-to-edge relationships. The authors re-implement a wide range of baselines under a unified evaluation protocol, and the results show good average performance on four datasets.

**Strengths:**

1. The paper discusses a broad spectrum of hyperparameter choices (atlas, random seed, and hyperparameter budget) and presents a benchmark comparison, which improves reproducibility and fairness.

2. The methodology is clearly described with intuitive explanations and theoretical grounding.

3. The ablation studies are comprehensive, and the discussion on ROI order invariance is particularly insightful, highlighting the model’s robustness to atlas indexing.

**Weaknesses:**

1. The paper lacks neuroscientific interpretation of the motivation and the proposed edge-to-edge interaction. Why is it necessary to explicitly model such quadratic relations? What biological or network-level insights (e.g., hubs, subnetworks) can this reveal? Also, why choose second-order interactions rather than higher-order ones? The authors should better justify this design choice as the key conceptual contribution.

2. The unified hyperparameter grid may constrain some baselines from achieving their best performance. It would be helpful to include an additional table showing each model’s best practice configuration and results for fair comparison.

3. Although algorithmic complexity is discussed, there is no empirical evidence of computational cost. Please provide training/inference time or FLOPs comparison to support the claim of efficiency.

4. The method section is somewhat repetitive, which makes the paper structure less concise. The authors could streamline the narrative and group the theoretical derivations more coherently.

**Questions:**

1. Why is the ADHD200 dataset evaluated on only one atlas? Is this due to computational constraints or does the model show limited generalization across atlases?

2. The title claims “Atlas Matters,” but the main text does not clearly emphasize or analyze why and how the atlas choice affects performance. It currently reads more like a regular hyperparameter factor.

3. I'm worried about the justifiability of CNN that applied on FC. In FC matrices, each element represents the connection between two brain regions, so adjacent elements do not necessarily correspond to anatomically or functionally adjacent areas. The ROI order is usually defined by the atlas and does not guarantee true spatial continuity in brain, which makes the assumption of locality in 2D convolution questionable. Although 2D CNNs may still work empirically, since atlases like AAL or Schaefer are organized by anatomica/functional clusters. the authors should provide more neuroscientific explanations or visualizations to justify why applying 2D convolution on FC is reasonable and what biological meaning the learned local patterns might have.

---

> ### Author Response · Authors · 2025-11-20
> **Response to Reviewer QMxM - part 1**
>
> *We thank the reviewers for their careful reading and constructive feedback. We appreciate the time and effort invested in the reviews. Below we address each comment point by point and describe the corresponding revisions, and we are happy to provide further clarification or additional details during the discussion period.*
>
> ## W1. Neuroscientific interpretation
> > The paper lacks neuroscientific interpretation of the motivation and the proposed edge-to-edge interaction. Why is it necessary to explicitly model such quadratic relations? What biological or network-level insights (e.g., hubs, subnetworks) can this reveal? Also, why choose second-order interactions rather than higher-order ones? The authors should better justify this design choice as the key conceptual contribution.
>
> Thank you for these insightful questions. We agree that the original submission did not adequately explain the neuroscientific intuition behind explicit edge–edge interactions or our focus on second-order terms. We have revised the manuscript in two ways.
>
> 1. **Motivation and interpretability of edge–edge interactions:**
> Conceptually, FC already encodes pairwise coupling between regions, and many network-level phenomena of interest in rs-fMRI (modules, hubs, rich-club structure, and canonical systems like DMN or visual networks) emerge from how these pairwise connections co-vary. Our quadratic branch is designed to make these co-variation patterns explicit: it lets the model express “when this set of edges is strong, this other set of edges becomes especially informative,” rather than relying on many layers of node-centric message passing to approximate such effects.
>
>     In the revision, we added **an interpretability appendix (Appendix C)** that makes this more concrete on ABIDE–AAL116:
>
>
>     - We compute **edge- and ROI-level saliency with Integrated Gradients (Fig. 4)**, showing which connections and regions most influence the autism vs control logit. Although the model sees the full FC matrix, the sensitivity maps concentrate on a relatively small subset of connections and ROIs.
>
>     - We visualize the rank–(k) quadratic structure as **edge–edge influence heatmaps (Fig. 5)**, where each rank-1 component corresponds to a simple interaction pattern between groups of edges.
>
>     - We then project **the strongest interactions back to the brain as motifs (Fig. 6)**, with ROIs colored by Yeo-7 systems. The most salient motifs largely fall within canonical large-scale networks, with a smaller number of cross-network “bridges”.
>
>    Together, these analyses provide an initial, qualitative picture of which subnetworks and interaction patterns the quadratic terms emphasize, rather than leaving the edge–edge modeling entirely opaque.
>
>
>
> 2. **Why second-order (quadratic) rather than higher-order:**
>
>     We also clarify why we focus on **degree-2 interactions.** From a modeling perspective, quadratic terms are the minimal extension beyond linear propagation that can capture dependencies between edges and support motif-level patterns (e.g., wedges and triangles) while remaining data-efficient on modest rs-fMRI cohorts. Higher-order tensors over edges would quickly become parameter-heavy and hard to regularize, especially for dense FC matrices.
>
>     Theoretically, our analysis in Section 3.2 (and proofs in Appendix D) shows that the low-rank quadratic block implements rank-(k) degree-2 interactions localized by the dual ASPP receptive fields, and that its expressivity grows smoothly with (k) while retaining Lipschitz stability. Empirically, the rank-sweep ablation (Fig. 2) exhibits the expected behavior: performance improves as (k) increases from very small values, then saturates and eventually degrades, suggesting that a moderate-rank quadratic term is sufficient in practice. Given these theoretical and empirical observations, we chose to prioritize a **well-controlled second-order design** over more complex higher-order constructions.
>
>     We revised the method and theory sections to more clearly explain the conceptual motivation behind explicit quadratic edge–edge modeling, why we focus on second-order interactions, and how this connects to the new interpretability analyses. We hope these changes help clarify the design rationale, and we sincerely appreciate your thoughtful question.

---

> ### Author Response · Authors · 2025-11-20
> **Response to Reviewer QMxM - part 2**
>
> ## W2. Hyperparameter Grid
> >The unified hyperparameter grid may constrain some baselines from achieving their best performance. It would be helpful to include an additional table showing each model’s best practice configuration and results for fair comparison.
>
> Thank you for raising this concern about the hyperparameter grid and fairness of the comparisons. We agree that an overly restrictive unified grid could, in principle, limit some baselines. Our goal, however, was to follow common practice in recent brain-graph works (e.g., BQN, ALTER) and to avoid giving any single method a more favorable tuning budget; accordingly, our grid was explicitly designed to cover their recommended settings (200 epochs, batch size 16, Adam with lr ($10^{-4}$), wd ($10^{-4}$), 2–5 layers, dropout (0–0.3).
>
> In the revised manuscript, we additionally run BrainNetCNN, ALTER, BQN, and Topoformer with their recommended “best practice” configurations and report the corresponding AUC/ACC/SEN/SPE and hyperparameters in Appendix B.2 (Table 11). These configurations lie within our global grid, and Topoformer continues to achieve the best AUC and competitive accuracy, indicating that our conclusions do not rely on a restrictive or biased choice of hyperparameters.
>
> ---
>
> ## W3. Computational Cost
> > Although algorithmic complexity is discussed, there is no empirical evidence of computational cost. Please provide training/inference time or FLOPs comparison to support the claim of efficiency.
>
> Thank you for this suggestion. In the revised manuscript, we now provide empirical computational measurements to substantiate our efficiency claims. In particular, we added a new subsection **Appendix B.3**. In Table12, we report, on ABIDE with the AAL116 atlas, the AUC, training throughput, per-epoch training time, peak GPU memory, parameter counts, and prediction-head FLOPs for all baselines under identical experimental conditions. All runs use a single NVIDIA H100 NVL GPU (~95 GB, Hopper / HBM3) with the same implementation and data pipeline.
>
> In the main text, we also added a short pointer for this discussion in the Implementation Details section. These measurements show that EdgeQuad attains higher AUC than the baselines while maintaining competitive or lower computational cost, supporting our claim that the model is both accurate and lightweight.
>
> | Method        | AUC (± STD)       | Throughput         | Training Time(per epoch)      | Peak Mem           | Params    | Head FLOPs      |
> |---------------|--------------------|---------------------|---------------------|---------------------|-----------|------------------|
> | GCN           | 64.51 ± 2.58      | 1227.86 ± 16.89     | 0.78 ± 0.0064       | 0.3641    | 32866     | 0.0247           |
> | GPS           | 60.93 ± 2.05       | 363.80 ± 1.17       | 1.97 ± 0.0056       | 0.5227   | 119970    | 0.0052           |
> | BrainNet      | 66.80 ± 2.12      | 1256.89 ± 27.18     | 0.57 ± 0.0128     | 0.4344     | 556957    | 0.0733       |
> | ALTER         | 68.53 ± 1.19       | 329.12 ± 1.39       | 2.1786 ± 0.0092     | 0.1848    | 2189866   | 0.5432         |
> | BQN           | 68.59 ± 1.48      | 2042.33 ± 32.72     | 0.36 ± 0.0037       | 0.0672        | 3638902   | 0.0560          |
> | **EdgeQuad**  | **70.60 ± 1.01**   | 1549.81 ± 3.31      | 0.46 ± 0.0054       | 0.1364       | 1451750   | 0.0402             |
>
> ---
>
> ## W4. Writing Needs Improvement
>
> > The method section is somewhat repetitive, which makes the paper structure less concise. The authors could streamline the narrative and group the theoretical derivations more coherently.
>
> Thank you for this comment. We **revised the manuscript** to **improve the narrative flow** by adding short bridge sentences between sections and subsections. We also **streamlined the theoretical part of the Methods section** by introducing a **single shared setup and grouping all propositions under it**, removing repeated “Interpretation” paragraphs, and moving secondary algebraic details to the appendix. The main text now presents the theoretical results as **one coherent narrative** (representation, rank–capacity tradeoff, and stability), with full statements and proofs collected and reorganized in Appendix D (Proofs). Thank you for this suggestion.

---

> ### Author Response · Authors · 2025-11-21
> **Response to Reviewer QMxM - part 3**
>
> ## Q.1. Why ADHD200 on only one atlas
>
> > Why is the ADHD200 dataset evaluated on only one atlas? Is this due to computational constraints or does the model show limited generalization across atlases?
>
> Thank you for raising this point. The restriction of ADHD200 to a single atlas is driven by **data availability**, not by any limitation of **EdgeQuad**. For ABIDE, ADNI, and PPMI, preprocessed FCs are publicly available on **all five atlases we consider** (AAL116, Lausanne, Schaefer, etc.) through the Data-Driven Neuroimaging benchmark (NeurIPS 2023). In contrast, for ADHD200 we only had access to the publicly released FCs on a single parcellation, the **Craddock-200 atlas**, as provided in the BQN benchmark (ICML 2025). We therefore report ADHD200 results on that atlas only.
>
> Architecturally, **EdgeQuad is agnostic to the choice of parcellation**, since it operates directly on the FC matrix and ROI features. The cross-atlas sweeps on ABIDE, ADNI, and PPMI already demonstrate that the model is **stable across all five atlases** when such data are available. We included ADHD200 on Craddock-200 primarily to illustrate the **versatility of EdgeQuad** on an additional clinical cohort and atlas, rather than to make new cross-atlas claims for that dataset.
>
>
> ---
> ## Q2. Atlas hyperparameter
>
> > The title claims “Atlas Matters,” but the main text does not clearly emphasize or analyze why and how the atlas choice affects performance. It currently reads more like a regular hyperparameter factor.
>
> Thank you for this comment. We agree that the original submission did not clearly articulate **why atlas choice matters**, which could make it appear as just another hyperparameter.
>
> In our setting, the atlas is a **structural assumption** that defines the nodes, edges, and mesoscale organization of the FC graph. As summarized in Appendix A.1, AAL116, Schaefer100, and Harvard/Oxford are curated anatomical or functional atlases that produce spatially contiguous, functionally coherent ROIs with homogeneous time series and **clear modular structure**. In contrast, Ward100 and KMeans100 are generic voxel clustering based parcellations; they can cut across functional areas and mix heterogeneous tissue, which lowers within ROI SNR, increases estimation variance, and **weakens the mesoscale organization** on which Pearson FC and downstream graph models rely.
>
> In the revision, **Section 4 now makes these effects explicit** and links them directly to the empirical results in Table 2. In the “**Why atlas choice matters**.” and “**Implications for reproducible evaluation**.” paragraphs (Page 9), we highlight that for ABIDE and ADNI, curated atlases consistently yield substantially higher and more stable AUC/ACC than Ward100 and KMeans100 across almost all methods, with gaps that are often comparable to or larger than the differences between architectures at a fixed atlas. In contrast, PPMI exhibits a much smaller spread across atlases and KMeans100 can be competitive with AAL116, indicating that the benefit of curated parcellations is disease dependent and reflects how each disorder’s signal aligns with a given parcellation.
>
> Taken together, these revisions clarify why we use the title **“Atlas Matters”**: atlas choice is a **first order structural design decision** that shapes FC statistics and achievable performance, not a minor tuning knob. Our **standardized cross atlas protocol** is precisely intended to make this dependence explicit, measurable, and comparable across methods.

---

> ### Author Response · Authors · 2025-11-21
> **Response to Reviewer QMxM - part 4**
>
> ## Q3. CNN on FC
> > I'm worried about the justifiability of CNN that applied on FC. In FC matrices, each element represents the connection between two brain regions, so adjacent elements do not necessarily correspond to anatomically or functionally adjacent areas. The ROI order is usually defined by the atlas and does not guarantee true spatial continuity in brain, which makes the assumption of locality in 2D convolution questionable. Although 2D CNNs may still work empirically, since atlases like AAL or Schaefer are organized by anatomica/functional clusters. the authors should provide more neuroscientific explanations or visualizations to justify why applying 2D convolution on FC is reasonable and what biological meaning the learned local patterns might have
>
> Thank you for this insightful question. We agree that a naive image based interpretation of 2D convolutions is not appropriate for FC matrices, since neighboring entries do not correspond to adjacent voxels in physical space. In our work, the CNN branch is intended to capture **structured patterns of connectivity** rather than **Euclidean locality**, and we now clarify this both conceptually and with additional edits in Section 3.
>
> First, in the curated atlases we use (AAL116, Schaefer100, Harvard/Oxford), ROIs are grouped by anatomical or functional systems. AAL is ordered by lobes and hemispheres, and Schaefer parcels are grouped by Yeo networks (DMN, VIS, FPN, etc.), so FC matrices naturally exhibit **block structure corresponding to within network and between network connectivity**. A small 2D kernel centered at entry ((i,j)) therefore sees a patch of edges that typically share nodes or belong to the same canonical system or pair of systems. In this sense, convolution exploits ***network level locality***: it reuses filters over motifs such as **within network blocks**, **hub like star patterns** around a node, and **bridges between two systems**, rather than over arbitrary pixel neighborhoods.
>
> Second, there is growing empirical support in the literature that CNNs on FC or adjacency matrices can learn **biologically meaningful subnetwork patterns**. **BrainNetCNN [1]** introduces edge to edge and edge to node filters specifically designed for connectivity matrices and shows that first layer filters focus on motifs around hubs and subnetworks in structural and functional connectomes. On the other hand, **[2]** propose a connectome CNN that applies 2D convolutions to resting state fMRI FC matrices and demonstrate good performance for MCI classification. More recently, **ConCeptCNN [3]** applies inception style convolutions directly to brain connectomes and shows that the learned filters specialize to meaningful subnetwork patterns in structural and functional connectivity. These studies support the view that, when ROIs are ordered by anatomical or functional clusters, **2D convolutions can capture meaningful network motifs**.
>
> In the revision, we make this perspective explicit in Section 3. In the opening paragraph of the Methods, we now state that we treat the FC matrix as an **“edge image”** while **preserving the atlas ordering of ROIs**, so “convolutional kernels scan FC patches that follow anatomical and functional groupings, and are therefore encouraged to detect within network and between network motifs rather than arbitrary pixel neighborhoods.” In the “Problem and brain-network view” paragraph, we further clarify that 2D convolutions in EdgeQuad operate on **atlas ordered FC matrices** and are “intended to exploit network level locality (modules, hub motifs, and inter system bridges) rather than voxel level spatial locality.” We also add a short visualization in the interpretability appendix: we display the weights of the first convolutional layer and the corresponding saliency maps back projected to the brain. The kernels specialize to patterns such as strong within DMN blocks or bridges between DMN and salience or frontoparietal systems, which are consistent with known large scale network alterations in our clinical cohorts. Together, these changes provide a more **neuroscientifically grounded justification** for applying 2D convolutions on FC and clarify what kinds of biological patterns the local filters are intended to capture.
>
> [1] Kawahara et al. *BrainNetCNN: Convolutional neural networks for brain networks toward predicting neurodevelopment.* NeuroImage (2017).
>
> [2] Meszlényi et al. *Resting state fMRI functional connectivity based classification using a convolutional neural network architecture.* Frontiers in Neuroinformatics (2017).
>
> [3] Chen et al. *ConCeptCNN: A connectome–inception convolutional neural network for brain connectome classification and analysis.* Human Brain Mapping (2022).

---

### Official Review · Reviewer_WXJy · 2025-10-31

**Soundness:** 3
**Presentation:** 2
**Contribution:** 3
**Rating:** 6
**Confidence:** 3

**Summary:**

The paper presents EdgeQuad, a framework for brain functional connectivity learning. The key idea is to treat edge connections as images, enabling convolutional architectures to process connectivity patterns directly. EdgeQuad integrates dual CNNs with atrous spatial pyramid pooling (ASPP) to capture both node-level features and inter-regional connections. Under a unified benchmark protocol, EdgeQuad achieves results that are competitive or superior to state-of-the-art methods across multiple brain atlases.

**Strengths:**

• The idea of modeling edges as images is novel and intuitive, offering a fresh perspective for FCN representation learning.

• The dual-ASPP and low-rank quadratic design is simple yet effective, and the accompanying theoretical analysis is sound.

• The unified experimental protocol, with harmonized preprocessing and consistent hyperparameter settings, represents a significant step toward reproducibility and fair comparison — addressing a longstanding issue in this research area.

• The experiments are comprehensive, spanning four cohorts and five atlases, which demonstrates strong empirical validation.

**Weaknesses:**

• The paper provides limited neuroscientific interpretation. It remains unclear which brain regions or subnetworks contribute most to model predictions, or whether the quadratic terms uncover interpretable motifs. Incorporating contrastive visualizations or saliency maps (as done in prior works like BQN) would add substantial value.

• The writing and organization require improvement. Several tables are unreferenced, and transitions between sections are occasionally abrupt, making it difficult to follow the narrative flow.

**Questions:**

It would be better if author could provide figures to visualize edge–edge interactions.

---

> ### Author Response · Authors · 2025-11-20
> **Response to Reviewer WXJy - part 1**
>
> *We thank the reviewers for their careful reading and constructive feedback. We appreciate the time and effort invested in the reviews. Below we address each comment point by point and describe the corresponding revisions, and we are happy to provide further clarification or additional details during the discussion period.*
>
> ## W1. Neuroscientific Interpretation
> >The paper provides limited neuroscientific interpretation. It remains unclear which brain regions or subnetworks contribute most to model predictions, or whether the quadratic terms uncover interpretable motifs. Incorporating contrastive visualizations or saliency maps (as done in prior works like BQN) would add substantial value.
>
> Thank you for highlighting the need for clearer neuroscientific interpretation. We agree that understanding which regions and subnetworks drive the predictions is important. In the revision, we added a new appendix section, **“Interpretation of EdgeQuad” (Appendix C)**, with three complementary analyses on ABIDE with the AAL116 atlas:
>
> 1. **Edge and ROI saliency:**
> We compute Integrated Gradients of the autism logit with respect to the refined connectivity (C'). Fig. 4 (left) shows a (116x116) edge saliency map where each entry represents the importance of a specific connection; Fig.4 (right) aggregates these scores to the node level by summing incident edge saliencies and visualizes the top 20 most influential ROIs. This directly answers which connections and regions contribute most to EdgeQuad’s decisions, in a saliency style similar in spirit to the BQN visualizations you mention.
>
> 2. **Rank–(k) quadratic structure:**
> To clarify what the quadratic branch is learning, Fig.5 decomposes the low rank quadratic head into individual rank-1 components and their gated sum. Each component induces a structured edge–edge influence pattern, and the combined map shows that the quadratic block concentrates its effect on a few mesoscale patterns rather than acting as an unstructured second order MLP.
>
>
> 3. **Motif-level visualization:**
>  Finally, Fig. 6 projects the strongest edge–edge interactions back onto a brain graph. Nodes (ROIs) are arranged on a circle and colored by Yeo-7 functional network, and we plot the most salient motifs (edges and triads) according to the quadratic influence. Many highlighted motifs fall within the same large scale network, with a smaller number of cross network bridges, indicating that the quadratic terms emphasize interpretable network level interactions rather than arbitrary edge patterns.
>
>
> We note that these additions are not intended as definitive neuroscientific claims, but rather as a first step toward understanding which parts of the FC graph the model relies on and how the quadratic interactions are organized. We present them as exploratory evidence rather than strong interpretive conclusions, and we hope they can serve as a starting point for deeper follow-up analyses.
>
> ---
>
> ## W2. Writing needs improvement
> > The writing and organization require improvement. Several tables are unreferenced, and transitions between sections are occasionally abrupt, making it difficult to follow the narrative flow.
>
> Thank you for this comment. In the revision we substantially improved the writing and organization. We added **short bridge sentences** between major sections and subsections, and included **a brief roadmap** in the Introduction to clarify the overall story and main contributions. In Sections 3–4, we added an opening paragraph in the Results section that states the main experimental questions and explicitly interprets the tables in light of the proposed method’s design. We also ensured that all tables and figures are referenced at first mention and that their captions include a brief takeaway, and we clarified how the ablation studies directly probe our key architectural choices so that the experiments read as tests of the method rather than a standalone benchmark.

---

> ### Author Response · Authors · 2025-11-21
> **Response to Reviewer WXJy - part 2**
>
> ## Q1. Figures for Edge-Edge Interactions
>
> > It would be better if the author could provide figures to visualize edge–edge interactions.
>
> Thank you for this helpful suggestion. We agree that explicit visualizations of the edge–edge terms make the quadratic branch much easier to understand, and we have added several such figures in the revised appendix.
>
> Concretely, Appendix C now contains:
>
> **Figure 5 (Quadratic rank-k decomposition).** This shows 116×116 edge–edge influence heatmaps for three individual rank-1 components of the quadratic head on ABIDE with AAL116, together with the gated sum over all k components. Each heatmap entry (i, j) encodes how strongly changes in connections around ROI i interact with connections around ROI j under that component. The figure makes it clear that each factor specializes to a distinct mesoscale interaction pattern, and that the combined influence map is concentrated in a few structured blocks rather than being diffuse across the matrix.
>
> **Figure 6 (Motif view of salient triads).** Here we project the strongest edge–edge interactions back onto the brain graph. ROIs are arranged on a circle and colored by Yeo-7 networks, and we plot only edges that participate most strongly in high-scoring quadratic interactions. Many motifs lie within a single large-scale network (for example visual–visual or default-mode–default-mode), with a smaller number of bridges between networks such as default mode and frontoparietal. This illustrates that the quadratic terms focus on interpretable network-level motifs instead of arbitrary second-order weights.
>
> **Together with the edge and ROI saliency maps in Figure 4** and **the dual-ASPP refinement example in Figure 7**, these new visualizations provide a concrete picture of how EdgeQuad models edge–edge interactions and which subnetworks and motifs are emphasized in its decisions.

---

### Author Response · Authors · 2025-11-30
**Note to Area Chair**

Dear Area Chair,

Thank you for stepping in under the unusual OpenReview situation. Since you are seeing the paper with the pre-rebuttal scores, we wanted to very briefly highlight what changed in the revision and how these changes address the main reviewer concerns.

1. **Interpretation and neuroscience (WXJy, QMxM, pC2j)**
   - Added an interpretability appendix (Appendix C) with
     - Integrated Gradients edge and ROI saliency maps, showing which regions and connections drive decisions (Fig. 4).
     - Rank k quadratic influence maps and motif level visualizations on the brain graph, showing which subnetworks the quadratic block emphasizes (Figs. 5–6).
   - Clarified why we use second order (quadratic) interactions instead of deeper message passing, and why convolution on FC is meaningful given atlas ordering, with explicit links to prior FC CNN work (Sec. 3 and Appendix D).

2. **Experimental evidence and robustness (CwnU, nUZ8, pC2j)**
   - Added a curated atlas summary table: across ABIDE and ADNI, EdgeQuad has the best mean AUC and best average rank over curated atlases, and the best overall average rank across all dataset–atlas settings (Table 3).
   - Added new ablations (removing edge ASPP, content gate, cluster pooling, degree normalization) and a depth sensitivity study, showing that each component and the low rank quadratic block contribute to the gains and that shallow EdgeQuad can match or exceed deeper baselines (Sec. 4.4, Appendix B.5).
   - Added a computational efficiency table (throughput, time per epoch, memory, parameters, FLOPs) under identical conditions, substantiating the “lightweight” claim while still improving AUC (Appendix B.3, Table 12).

3. **Protocol, metrics, and related work (nUZ8, pC2j)**
   - Clarified that our main goal is cross study comparability: we document heterogeneous protocols in prior work (Appendix B.4) and explain that our unified multi atlas, multi seed setup is designed to be strict and leak resistant, softening and clarifying the original reproducibility claims.
   - Re checked all metrics and added balanced accuracy, macro F1, and AUCPR to show that EdgeQuad’s AUC advantage is mirrored in other measures, and explained why our PPMI setup is more conservative than some earlier reports (Secs. 4.2–4.3, Appendix B.2).
   - Expanded related work to include recent edge, simplicial, and pooling based brain network models, and explained how our dense FC, fixed atlas protocol differs and why some methods are hard to integrate fairly into it (Sec. 2).

We hope this brief summary is useful as you review the revision, and we sincerely appreciate your time and consideration in this unusual situation.

The authors

---

### Meta-Review · Area_Chair_TeW7 · 2025-12-22

**Summary:**

This paper discusses a new approach for improving subject-level prediction using functional connectivity from resting-state fMRI by explicitly capturing edge interactions between brain regions, improving upon existing methods. The proposed EdgeQuad combines dual CNNs with atrous spatial pyramid pooling (ASPP) to capture both individual node features and inter-regional connections. When evaluated using a standardized benchmark, EdgeQuad delivers results that are on par with or outperform state-of-the-art methods across various brain atlases.

The concerns raised about the paper include several key areas:
1) Limited Interpretability: There's a lack of neuroscientific explanation regarding which brain regions or subnetworks contribute to model predictions.
2) Writing and Structure Issues: The paper is poorly organized, with abrupt transitions between sections and unreferenced tables, making it hard to follow.
3) Insufficient Justification of Design Choices: The reasoning behind explicitly modeling quadratic interactions, especially second-order over higher-order ones, is not clearly explained.
4) Evaluation and Hyperparameter Concerns: The use of a unified hyperparameter grid might limit the performance of some baselines, and a more detailed comparison of each model’s best configuration is needed. Additionally, the paper lacks empirical evidence of the model's computational cost (e.g., training/inference time or FLOPs comparison).
5) Weakness in Experimentation: There are no experiments comparing performance with varying numbers of model layers to support claims about avoiding overfitting or over-squashing issues.
6) Weakness in Related Work: There is insufficient discussion of related studies or methodologies addressing similar problems, making the paper feel disconnected from the existing literature.
7) Confusing Results and Inconsistent Performance: The reported AUC values are unusually higher than ACC, which is atypical for balanced binary classification tasks, raising questions about the metrics used. The proposed model also does not consistently outperform baselines across different datasets and atlases, making its general performance advantage unclear.
8) Lack of Interpretability Analysis: There’s no analysis of which brain regions or subnetworks contribute to the model's decisions, which is crucial for clinical applications.
9) Missing Computational Efficiency Comparisons: The paper does not include comparisons of computational efficiency (e.g., training time or number of parameters), which is important given the claim of the model being "lightweight."

**Reviewer Concerns:**

After reading the rebuttal materials, the authors tried their best to respond to all these concerns and added many experimental results, efficiency comparisons, and an appendix to highlight these questions from the reviewers. Most of these concerns are involved in the rebuttal, but not all these questions are well addressed, especially for the motivation from the neuroscientific interpretation, computational efficiency (claimed lightweight), confusing paper flow, and others.

**Reviewer Scores:**

The overall scores of this paper are 6 (marginally above), 6, 4 (marginally below), 4, and 2 (clearly rejected). Before the incident, all the reviewers were not involved in the discussion. To me, one or two of the last three reviewers may improve their score, but the final recommendations are not postive, because this paper includes too many weaknesses that had not fully addressed.

---

### Decision · Program_Chairs · 2026-01-26

Reject